# Frontal eye field and caudate neurons make different contributions to reward-biased perceptual decisions

**Yunshu Fan, Joshua I Gold, Long Ding\***

Department of Neuroscience and Neuroscience Graduate Group, University of Pennsylvania, Philadelphia, United States

**Abstract** Many decisions require trade-offs between sensory evidence and internal preferences. Potential neural substrates include the frontal eye field (FEF) and caudate nucleus, but their distinct roles are not understood. Previously we showed that monkeys' decisions on a direction-discrimination task with asymmetric rewards reflected a biased accumulate-to-bound decision process (Fan et al., 2018) that was affected by caudate microstimulation (Doi et al., 2020). Here we compared single-neuron activity in FEF and caudate to each other and to accumulate-to-bound model predictions derived from behavior. Task-dependent neural modulations were similar in both regions. However, choice-selective neurons in FEF, but not caudate, encoded behaviorally derived biases in the accumulation process. Baseline activity in both regions was sensitive to reward context, but this sensitivity was not reliably associated with behavioral biases. These results imply distinct contributions of FEF and caudate neurons to reward-biased decision-making and put experimental constraints on the neural implementation of accumulation-to-bound-like computations.

**\*For correspondence:**
lding@pennmedicine.upenn.edu

## Introduction

Complex decisions often require interpreting external sensory inputs in the context of outcome expectations and preferences. This kind of decision-making is pervasive in our daily lives, balancing what we observe with what we desire. Under controlled task conditions, both humans and non-human animals tend to achieve this balance in a roughly normative manner. Specifically, when the sensory evidence strongly supports a particular option, decision-makers tend to choose that option independent of alternative expectations and preferences. Conversely, when the sensory evidence is weak, decision-makers tend to make more and faster choices to options with preferred, expected outcomes (*Maddox and Bohil, 1998*; *Voss et al., 2004*; *Diederich and Busemeyer, 2006*; *Liston and Stone, 2008*; *Whiteley and Sahani, 2008*; *Feng et al., 2009*; *Summerfield and Koechlin, 2010*; *Teichert and Ferrera, 2010*; *Gao et al., 2011*; *Leite, 2012*; *Mulder et al., 2012*; *Blank et al., 2013*; *Fan et al., 2018*; *Waiblinger et al., 2019*). However, exactly how and where in the brain the computations needed for these flexible decision processes are implemented is not well understood.

The observed patterns of choices and reaction times (RTs) for these kinds of tasks are often consistent with an accumulate-to-bound (drift-diffusion) decision process (*Ratcliff, 1978*; *Maddox and Bohil, 1998*; *Gold and Shadlen, 2002*; *Voss et al., 2004*; *Bogacz et al., 2006*; *Diederich and Busemeyer, 2006*; *Bogacz, 2007*; *Feng et al., 2009*; *Simen et al., 2009*; *Krajbich et al., 2010*; *Summerfield and Koechlin, 2010*; *Gao et al., 2011*; *Leite, 2012*; *Mulder et al., 2012*; *Blank et al., 2013*; *Fan et al., 2018*). Within this framework, asymmetric reward-choice associations (reward contexts) induce biases in the evidence-accumulation process, the decision bounds for different choice options, or both. The relative contributions of these different forms of reward context-dependent

bias likely reflect specific adaptive strategies and can vary by task design, subject, and testing days (*Fan et al., 2018*).

How does single-neuron activity relate to the computations required for incorporating reward and visual information to form decisions? Previously we trained monkeys to perform an asymmetric-reward direction-discrimination task (*Figure 1A*), in which the monkeys report their perception of the global motion direction of a noisy stimulus with eye movements under different reward contexts. We showed that activity of some caudate neurons was sensitive to both reward context and motion stimulus. In addition, caudate microstimulation affected the monkeys' reward biases in a manner that reflected coordinated changes in drift rates and relative bound heights in a drift-diffusion decision framework (*Doi et al., 2020*).

To provide additional insights into the neural implementation of these decision computations in multiple brain regions, we examined both the caudate nucleus and one of its major cortical input sources, the frontal eye field (FEF) of the lateral prefrontal cortex. Neurons in these two regions contribute to perceptual and reward-based decision making along with reward-modulated motor performance (for a very limited sample, see *Thompson et al., 1996*; *Thompson et al., 1997*; *Kawagoe et al., 1998*; *Kim and Shadlen, 1999*; *Freedman et al., 2001*; *Schall, 2001*; *Coe et al., 2002*; *Kobayashi et al., 2002*; *Lauwereyns et al., 2002b*; *Lauwereyns et al., 2002a*; *Roesch and Olson, 2003*; *Heekeren et al., 2004*; *Samejima et al., 2005*; *Ding and Hikosaka, 2006*; *Nakamura and Hikosaka, 2006a*; *Nakamura and Hikosaka, 2006b*; *Boettiger et al., 2007*; *Lau and Glimcher, 2007*; *Lau and Glimcher, 2008*; *Pan et al., 2008*; *Ferrera et al., 2009*; *Basten et al., 2010*; *Ding and Gold, 2010*; *Cai et al., 2011*; *Ding and Gold, 2012c*; *Ding and Gold, 2012a*; *Heitz and Schall, 2012*; *Seo et al., 2012*; *Kim and Hikosaka, 2013*; *Teichert et al., 2014*; *Yanike and Ferrera, 2014b*; *Ding, 2015*; *Hanks et al., 2015*; *Santacruz et al., 2017*; *Amemori et al., 2018*; *Schall, 2019*). Functional imaging and modeling studies also suggest that the two regions are involved in complex decisions that balance visual evidence and reward expectation to guide appropriate movements (*Rao, 2010*; *Summerfield and Koechlin, 2010*; *Chen et al., 2015*). However, their specific computational roles in these decisions, represented at the single-neuron level, remain largely speculative.

In this study, we focused on three questions: (1) How do FEF and caudate neurons encode key task factors including choice, motion strength, reward context, and reaction times (RT)? (2) Do any of these task-related modulations in either brain area reflect the monkeys' behaviorally derived biases in drift rates? (3) Do any of these task-related modulations in either brain area reflect the behaviorally derived biases in relative bound heights?

## Results

We recorded from 149 FEF neurons from three monkeys (*n* = 85, 24 and 40 from monkeys A, C and F, respectively) and, in separate sessions, from 140 caudate neurons from the same monkeys (*n* = 18, 49, and 73 from monkeys A, C and F, respectively) performing the asymmetric-reward direction-discrimination task. As we reported previously (*Fan et al., 2018*), the monkeys' choices and RTs tended to reflect the strength (coherence) and direction of the visual motion stimulus but with a bias toward the large-reward option (*Figure 1B,C*).

### Diverse task-relevant sensory and reward encoding in both brain regions

Individual neurons in both the FEF and caudate showed a diversity of task-driven responses (several examples are illustrated in *Figure 2*; population summaries are shown in *Figure 2—figure supplement 1* and *Figure 3*). The FEF neuron in *Figure 2A* responded to choice target presentation with phasic (transient) and tonic (sustained) activation, showed a dip in activity after motion onset, then had gradually increasing activity (more for trials resulting in a contralateral choice) during motion viewing until a saccade-related burst for the contraversive saccade and a return to baseline activity for the other saccade. The FEF neuron in *Figure 2B* was activated after target onset, with higher activation when the contralateral choice was paired with large reward (red curves > green curves).

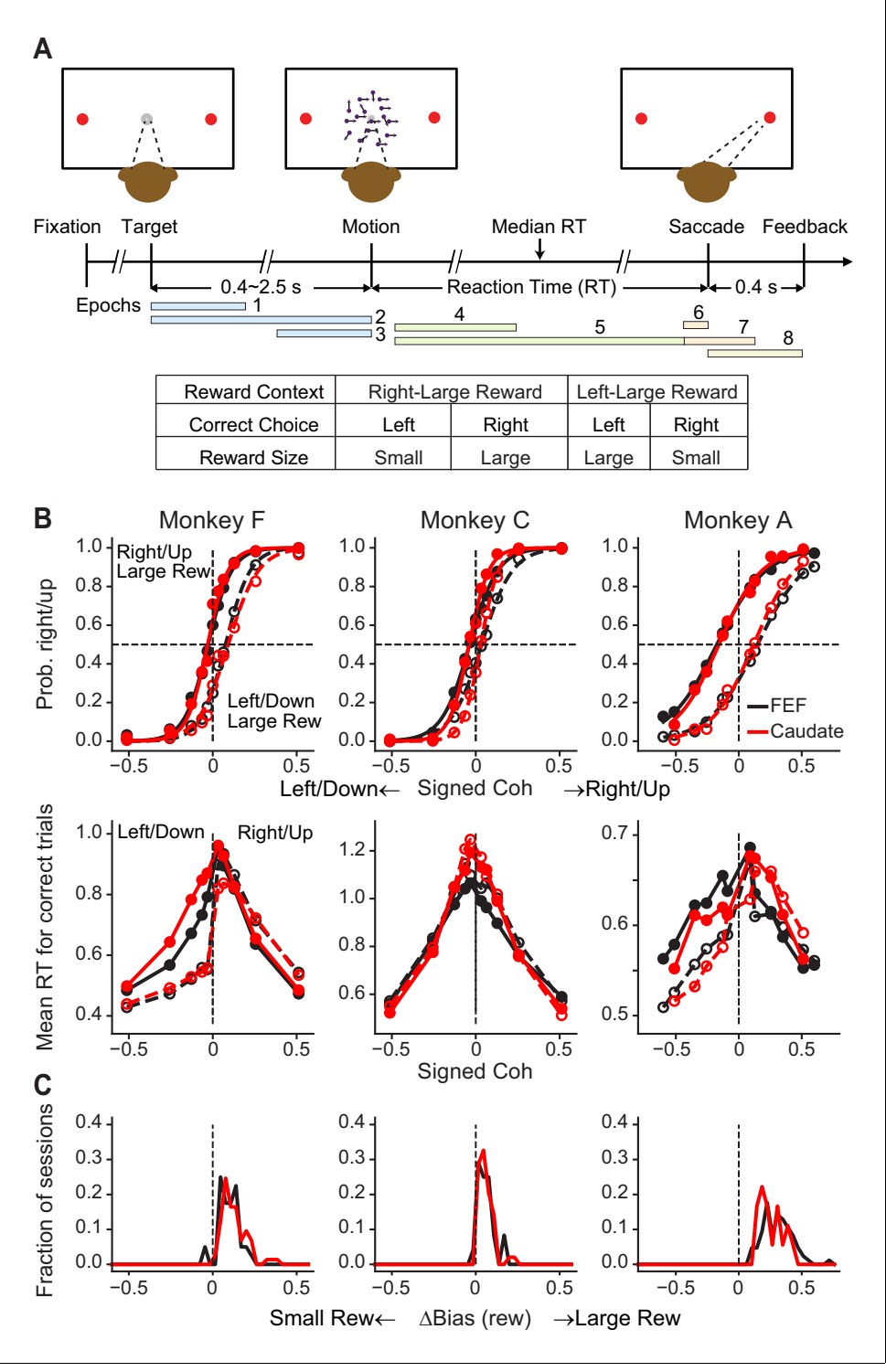

**Figure 1.** Monkeys biased toward choices associated with large reward. (**A**) Task design and timeline. Monkeys reported the perceived motion direction with a saccade to one of the two choice targets. The motion stimulus was turned off upon detection of the saccade. Correct trials were rewarded based on the reward context. Error trials were not rewarded. The color bars in the timeline indicate epoch definitions for the regression analysis of neural firing rates in *Equation 1*. (**B**) Average choice (top) and RT (bottom) behavior of three monkeys for sessions with FEF and caudate recordings. The FEF dataset (black) included 16,561 trials from 33 sessions for monkey F, 7924 trials from 23 sessions for monkey C, and 24,419 trials from 69 sessions for monkey A. The caudate dataset (red) included 26,614 trials from 69 sessions for monkey F, 21,076 trials from 44 sessions for monkey C, and 6309 trials
*Figure 1 continued on next page*

*Figure 1 continued*

from 17 sessions for monkey A. Filled and open circles: data from the two reward contexts. Similar results were reported previously for sessions with caudate recordings (*Doi et al., 2020*). (C) Histograms of reward bias for all sessions, estimated using logistic fits to choice data. Note that the bias magnitude varied in magnitude across monkeys and sessions, depending on the large:small reward ratio, the motion-coherence levels used in a given session, and the monkeys' inherent perceptual sensitivity (*Fan et al., 2018*).

This modulation by reward context persisted during a gradual ramp in activity during motion viewing (more for trials with the contralateral choice and for blocks when the contralateral choice was paired with large reward; *t*-test for $H_0$: regression coefficient for reward context = 0, p<0.05 for all epochs 1–8). This neuron also showed a saccade-related burst for the contraversive saccade. The FEF neuron in *Figure 2C* showed phasic activation by choice targets and motion onset, with activity that decreased during motion viewing, more gradually for the contralateral choice and higher coherences (compare curves with different shades), until reaching a saccade-related suppression.

The caudate neuron in *Figure 2D* did not respond to target onset but was activated after motion onset, with higher activity for the contralateral choice, at higher coherence, and in blocks when the contralateral choice was paired with large reward. These coherence and reward-context modulations persisted through saccade onset, with no convergence before saccade onset. The caudate neuron in *Figure 2E* also did not respond to target onset but was activated after motion onset for both choices, with a preference for ipsilateral choices that were paired with the small reward. After an initial large activation, this neuron gradually reduced firing toward saccade onset, largely maintaining reward-context and coherence modulation until after saccade onset.

These diverse trends were apparent across the populations of recorded FEF and caudate neurons. Most neurons in our sample had responses that were modulated, on average, over multiple time points during each trial, albeit with differences across neurons and brain areas. FEF neurons typically had elevated responses that began just after target onset and then persisted through motion viewing until the saccadic response (*Figure 2—figure supplement 1A*; for most neurons, spike rate increased just after target onset). Caudate neurons typically did not respond strongly to target onset but then had elevated responses during motion viewing and through the saccadic response (*Figure 2—figure supplement 1B*). Both regions included neurons with activity that increased and/or decreased relative to baseline at various time points during each trial.

To assess how these responses were modulated by choice, coherence, reward context, expected reward size for a given choice, and reaction time (RT), we first used linear regression applied to neural data in pre-defined task epochs in *Figure 1A* (*Figure 3A–D*). Because neurons in both brain areas represent a coherence- and time-dependent decision process (*Kim and Shadlen, 1999*; *Ding and Gold, 2010*; *Ding and Gold, 2012c*) that can conflate the effects of those two factors on neural responses, and because RT was modulated strongly by reward context in the current task, we conducted two sets of analyses: (1) using all of the factors listed above including coherence but not RT (*Figure 3A,B*), and (2) using all of the factors listed above including normalized RT (normalized separately for each reward context x choice combination) but not coherence (*Figure 3C,D*).

These epoch-based analyses showed several differences between the FEF and caudate populations, including: (1) a larger fraction of FEF neurons showed choice selectivity around and after saccade onset (*Figure 3A and C*, first column); (2) although selectivity for reward context emerged before motion onset in both populations and persisted through a trial, a larger fraction of caudate neurons showed such selectivity during motion viewing (second column); (3) a larger fraction of caudate neurons showed selectivity for reward size (third column); and (4) both populations showed significant coherence selectivity after motion onset, but a larger fraction of caudate neurons remained coherence-selective after a saccade was made (*Figure 3A*, fourth column). The caudate, but not FEF, population showed above-chance fractions of neurons with joint modulation by both reward and motion coherence (*Figure 3B*). Activity in both regions was related to the RT in similar fashions (*Figure 3C and D*). The RT-based regression also captured a larger variance of activity than the coherence-based regression in both populations (*t*-test on the explained variance, p<0.0001 and p=0.007 for FEF and caudate, respectively).

To examine these task-related modulations at a finer time resolution, we applied the RT-based linear regression to neural data in sliding windows (*Figure 3E–I*). These analyses produced results

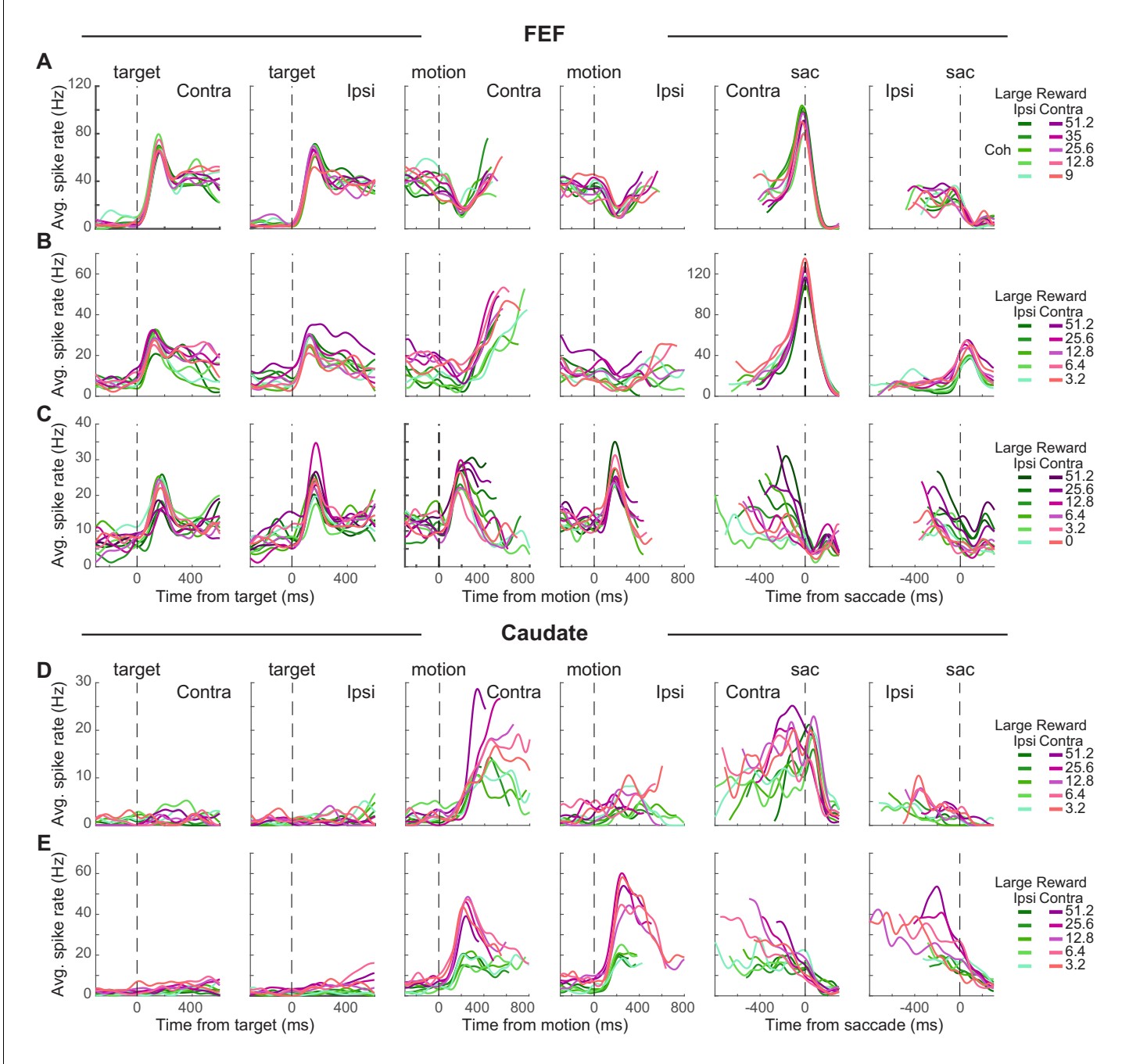

**Figure 2.** Task-related activity in FEF and caudate neurons. (**A-C**) Activity of three example FEF neurons. For display purposes, average spike count was measured for correct trials only and convolved with a Gaussian kernel (sd = 40 ms). Green colors: large reward was paired with the ipsilateral choice. Red colors: large reward was paired with the contralateral choice. Shades: coherence levels. For alignment to motion onset, activity was truncated at 100 ms before the median reaction time. For alignment to saccade onset, activity was truncated at 200 ms after the median time for motion onset. (**D-E**) Activity of two example caudate neurons. Same format as A.

The online version of this article includes the following figure supplement(s) for figure 2:

**Figure supplement 1.** Summary of activity patterns.

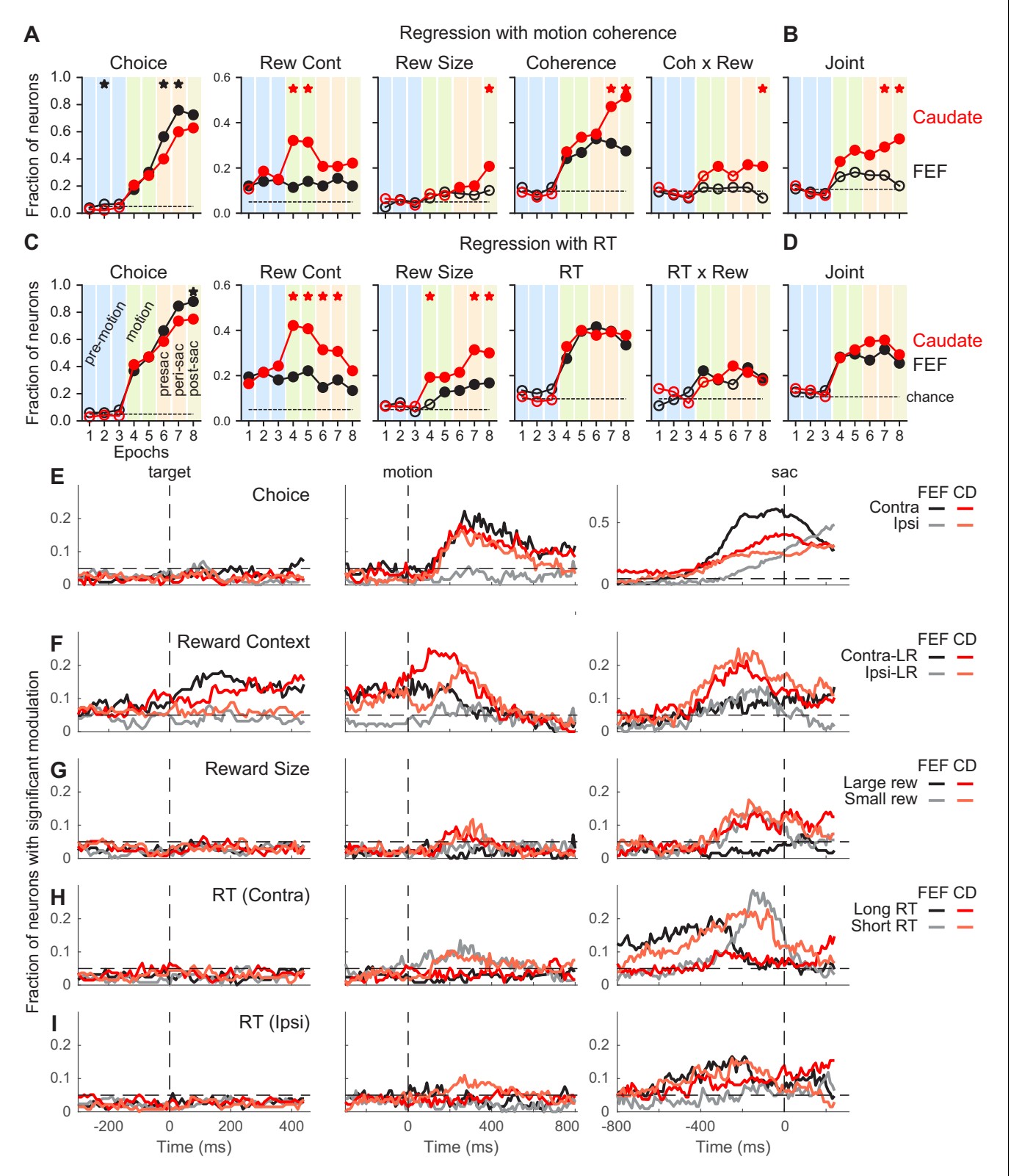

**Figure 3.** Comparison of task-related modulation of FEF and caudate activity. (**A**) Fractions of FEF (black) and caudate (red) neurons showing significant regression coefficients in the multiple linear regression in *Equation 2*. Criterion: t-test, p<0.05. Dashed lines: chance level, adjusted for the number of comparisons. Filled circles: the fraction was significantly greater than chance level (Chi-square test, p<0.05/72 (8 epochs x nine comparisons)). 'Coherence' and 'Coh x Rew': neurons with significant coefficients for either choice. Vertical color bars indicate epochs defined in *Figure 1A*. Stars

*Figure 3 continued on next page*

*Figure 3 continued*

indicate epochs in which the fractions differed between FEF and caudate populations (Chi-square test, p<0.05/72). (B) Fraction of neurons with joint modulation by coherence and reward-related terms. Same format as A. (C, D) Fractions of neurons showing significant regression coefficients in the multiple linear regression in *Equation 3*. Same format as A and B. (E-I) Fractions of neurons showing significant non-zero regression coefficients for different regressors (*Equation 3*). Results from RT-reward interaction terms were omitted because both regions showed near chance-level fractions. Dashed horizontal lines: chance level. Only neurons tested with non-vertical motion stimuli were included (n = 126 and 136 for FEF and caudate, respectively).

The online version of this article includes the following source data and figure supplement(s) for figure 3:

**Source data 1.** Source data for *Figure 3A–D*.
**Source data 2.** Source data for *Figure 3E–I*: FEF activity aligned to target onset.
**Source data 3.** Source data for *Figure 3E–I*: caudate activity aligned to target onset.
**Source data 4.** Source data for *Figure 3E–I*: FEF activity aligned to motion onset.
**Source data 5.** Source data for *Figure 3E–I*: caudate activity aligned to motion onset.
**Source data 6.** Source data for *Figure 3E–I*: FEF activity aligned to saccade onset.
**Source data 7.** Source data for *Figure 3E–I*: caudate activity aligned to saccade onset.
**Figure supplement 1.** Comparison of the time course of task-related modulation of FEF and caudate activity in monkey F.
**Figure supplement 2.** Comparison of the time course of task-related modulation of FEF and caudate activity in monkey C.
**Figure supplement 3.** Comparison of the time course of task-related modulation of FEF and caudate activity in monkey A.
**Figure supplement 4.** dPCA results for FEF activity aligned to motion onset.
**Figure supplement 5.** dPCA results for FEF activity aligned to saccade onset.
**Figure supplement 6.** dPCA results for caudate activity aligned to motion onset.
**Figure supplement 7.** dPCA results for caudate activity aligned to saccade onset.

that were consistent with the epoch-based analyses and showed further between-region differences in the timing and direction of task-related activity modulations. Selectivity for choice tended to increase during motion viewing until around the saccade, with stronger selectivity for contralateral/upward choices in both regions but particularly in the FEF (*Figure 3E*). Selectivity for reward context was evident before motion onset and continued toward saccade onset, with a mixture of preferences for the two reward contexts (*Figure 3F*). For FEF neurons, a higher fraction preferred the contralateral-Large Reward context before and during early motion viewing and similar fractions preferred either contexts before saccade onset. Although this was also true for the caudate population, the extent of the laterality was weaker. Selectivity for reward size, independent of the actual choice made, was most evident for data aligned to the saccade, with similar fractions of neurons of the caudate population preferring large or small reward (*Figure 3G*). Very few FEF neurons showed reward-size selectivity. Selectivity for RT was evident in both regions, with a dominant preference for short RTs associated with contralateral choices (*Figure 3H*) and mixed preferences otherwise (*Figure 3I*). These general patterns were present in the three monkeys for both FEF and caudate data (*Figure 3—figure supplements 1–3*).

To further characterize the modulation patterns, we applied the demixed principal component analysis (dPCA) method for the two populations (*Kobak et al., 2016*). Although our sample size was relatively small and trials were inherently unbalanced for different reward-choice-coherence combinations for this method, the dPCA results corroborated several findings from the multiple linear regression analysis (*Figure 3—figure supplements 4–7*), including: (1) choice-related components tended to account for a larger portion of variance in FEF activity than caudate activity (panels B and C in each figure, purple); (2) reward context-related components tended to account for a larger portion of variance in caudate activity than FEF activity (orange), particularly for around saccade onset; (3) coherence-related components tended to account for a larger portion of variance in FEF activity around motion onset than for activity around saccade onset, while the opposite was true for caudate activity (cyan); and (4) coherence and reward context or size interactions accounted for substantial variance for both regions (dark and light green). Collectively, these results indicated that neurons in both FEF and caudate represent a variety of task-relevant signals that could, in principle, support reward-biased perceptual decisions, but with different prevalences and preferences.

## Predictions of the biased decision variable in a drift-diffusion framework

As we showed previously, these monkeys' patterns of choices and RTs on this task were consistent with a drift-diffusion model (DDM; *Fan et al., 2018*). According to this model, a decision is formed when accumulated motion evidence reaches one of two pre-defined (collapsing) decision bounds (*Figure 4A*). The monkeys' reward-driven biases arose from coordinated, reward context-dependent adjustments of the rate of accumulation (drift rate, which scales with motion coherence) and relative bound heights (*Figure 4E*; for more details see *Fan et al., 2018*). A bias in the drift rate (ΔDrift, corresponding to the *me* parameter in the DDM) can be implemented as a constant offset to the momentary motion evidence (*Figure 4B*). A bias in the relative bound heights (ΔBound, corresponding to the *z* parameter in DDM) can be implemented as an offset in the starting value of the accumulation process (*Figure 4C*), an asymmetry in the absolute bound heights for the two choices (*Figure 4D*), or a combination of the two.

These different ways of implementing reward-driven biases correspond to different predictions of how and when the accumulating decision variable is modulated by reward context during a trial. Specifically, in the presence of a bias in the drift rate, the slope of the decision variable would be modulated by reward context during evidence viewing. In the presence of a bias in the starting value, the baseline value of the decision variable would be modulated by reward context before evidence onset. In the example in *Figure 4C*, the baseline value would be higher when the reward bias favors the upper-bound choice. In the presence of a bias in the bound heights, the decision variable would be modulated by reward context at the time of decision commitment. In the example in

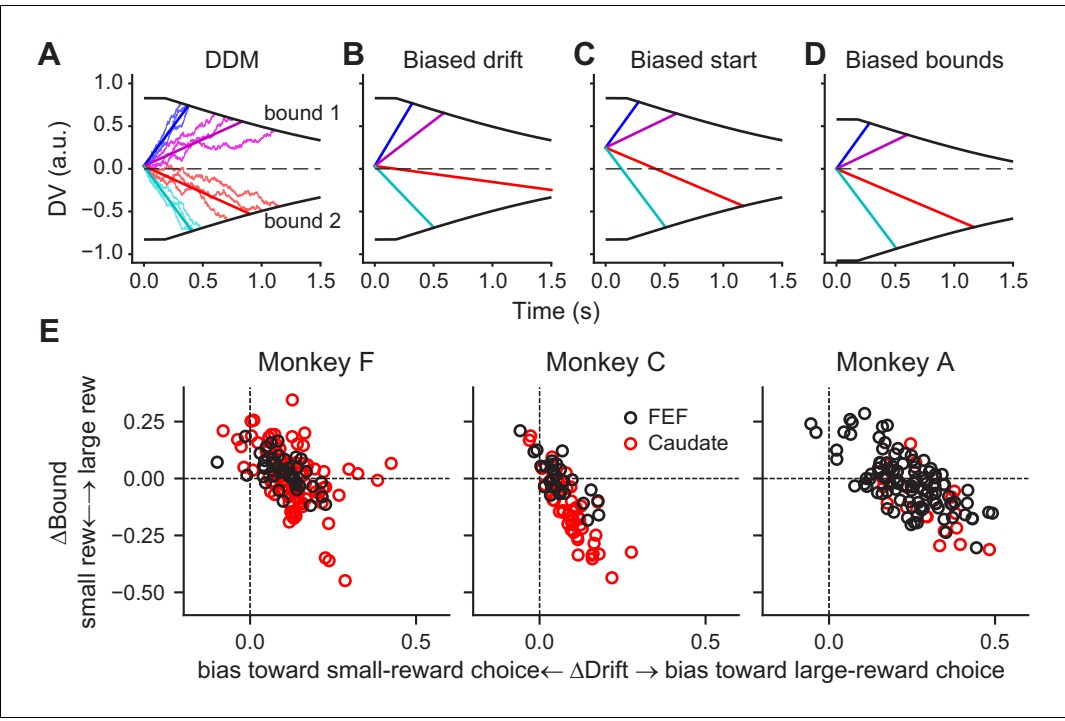

**Figure 4.** DDM illustration and fitted reward bias terms. (**A**) Drift-diffusion model (DDM). Evidence is accumulated over time into a decision variable (DV). A decision is made when DV crosses either collapsing bound. Thin noisy lines represent simulated DVs for two coherence levels and two motion directions (three trials for each combination). The straight lines represent the average DVs. (**B-D**) Illustration of different implementations of a bias favoring the upper-bound choice. (**B**) Drift rates are biased by adding a constant positive value to the evidence, resulting in steeper slopes for motion to the upper-bound choice and shallower slopes for motion to the lower-bound choice. (**C**) The accumulation begins with a positive starting value. (**D**) The accumulation ends at a lower absolute value for upper-bound choices than for lower-bound choices. (**E**) Summary of reward biases in drift and bound terms from DDM fits for the three monkeys. Positive values indicate biases toward the large-reward choice. Black and red data points represent sessions with FEF and caudate recordings, respectively.

*Figure 4D*, the ending value would be closer to the starting point of the evidence accumulation when the reward bias favors the upper-bound choice.

We examined whether FEF and caudate activity conform to these predictions. Given the asymmetric effects of these predicted biases on the two choices, we focused on neurons with reliable choice selectivity (*Figure 5*) and present results from other neurons as supplements when appropriate. The 'choice-selective' neurons were identified as showing significant and consistent choice modulation through motion viewing (epoch #5 in *Figure 1*) and before saccade onset (epoch #6), based on RT-based regression analysis results shown in *Figure 3*. The numbers of neurons meeting these criteria for the three monkeys are shown in *Table 1*. The average activity of choice-selective and

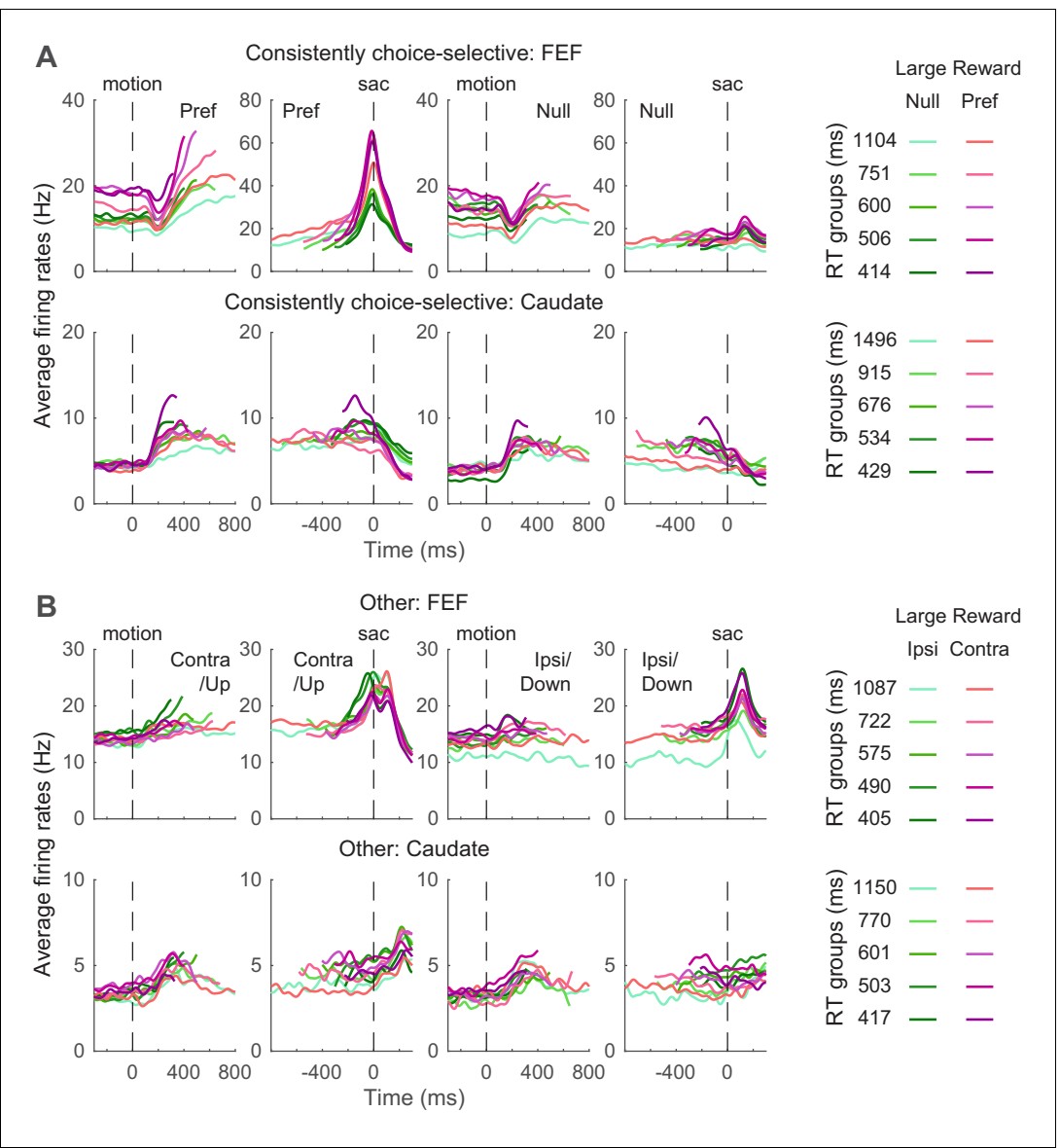

**Figure 5.** Average activity for neuron categories. (**A**) Average firing rates of neurons with significant and consistent choice selectivity. See *Table 1* for number of neurons in each category. Trials were grouped by choice (left and right rows), reward context (magenta/green), and RT quintiles (shade). Activity was aligned to motion and saccade onsets for the top and bottom rows, respectively. Only correct trials were included. For motion onset alignment, firing rates were truncated at the median RT minus 100 ms for each group. For saccade onset alignment, firing rates were truncated before median motion onset plus 200 ms for each group. For display purposes, firing rates were convolved with a Gaussian kernel (sigma = 25 ms). (**B**) Average firing rates of other neurons. Same format as A.

**Table 1.** Summary of counts/percentages for neurons with task-modulated activity.

| | FEF | | | Caudate | | |
|---|---|---|---|---|---|---|
| | Monkey A | Monkey C | Monkey F | Monkey A | Monkey C | Monkey F |
| Total | 85 | 24 | 40 | 18 | 49 | 73 |
| Consistently choice-selective | 35 | 14 | 7 | 6 | 31 | 21 |
| | 41% | 58% | 18% | 33% | 63% | 29% |
| Coherence-modulated slope of firing rate during motion viewing | 44 | 12 | 26 | 10 | 33 | 41 |
| | 52% | 50% | 65% | 56% | 67% | 56% |
| Reward context-modulated slope of firing rate during motion viewing | 21 | 5 | 3 | 2 | 9 | 15 |
| | 25% | 21% | 8% | 11% | 18% | 21% |
| Reward context-modulated activity before motion onset | 38 | 12 | 19 | 15 | 34 | 39 |
| | 45% | 50% | 48% | 83% | 69% | 53% |
| Reward context-modulated activity just before saccade onset | 51 | 16 | 26 | 15 | 34 | 49 |
| | 60% | 67% | 65% | 83% | 69% | 67% |

other neurons is shown in *Figure 5*. Note that because different coherence levels were used for the three monkeys, we grouped the trials by quintiles of RT for these plots.

Even using these common criteria for categorization, the average activity patterns of neurons in the same category differed for FEF and caudate. First, consider the choice-selective subpopulations. Whereas choice-selective FEF activity appeared to be roughly consistent with bound-crossing in the DDM (i.e., reaching a fixed level of activity at the end of the decision process and before saccade onset, regardless of the time it took to reach the decision), choice-selective caudate activity did not (*Ding and Gold, 2010*; *Ding and Gold, 2012b*). For trials in which the monkey made the preferred choice of the given neuron, the slope of FEF activity during motion viewing appeared to show more separations than the slope of caudate activity, between reward contexts and RT groups. The baseline FEF activity before motion onset appeared to differ more between reward contexts. The perisaccade activity for the preferred choice appeared to show opposite selectivity for reward context in the two regions (the purple curves tended to be above and below the green curves for FEF and caudate, respectively).

Second, in the other subpopulations that did not exhibit consistent choice selectivity, the average caudate activity appeared to maintain RT separation through saccade generation and onward, whereas the average FEF activity appeared to converge around saccade onset (*Figure 5B*). These apparent differences suggest that activity in the two regions may relate differently to the predictions of the DDM, which we examine in more detail below.

## FEF activity reflected behaviorally derived reward-driven drift-rate biases

We first examined whether FEF and caudate activity reflected evidence accumulation with a reward-driven bias in the drift rate. As illustrated in *Figure 4B*, such a signal is expected to show two features in neural activity. First, the rate of accumulation depends on motion coherence. For individual neurons, this dependence translates to motion-coherence modulation of the slope of firing rates during motion viewing. *Figure 6A and B* illustrate our procedure for estimating the slope of change in firing rates and its modulation by coherence, reward context, and their interaction. Second, the reward-context modulation of the slope of change reflects the behavioral reward bias in drift rate. In the model, the reward bias in drift rates is independent of coherence and the drift-rate scaling and dependent only on reward context. The corresponding modulation of the (slope of) activity of individual neurons is thus by reward context alone and not by the reward context-coherence interaction. Because neurons showed substantial variations in their firing-rate ranges, we used each neuron's modulation by coherence to normalize its modulation by reward context. The second expectation thus translates to a correlation between this normalized quantity and the behaviorally estimated bias in drift rate across neurons/sessions.

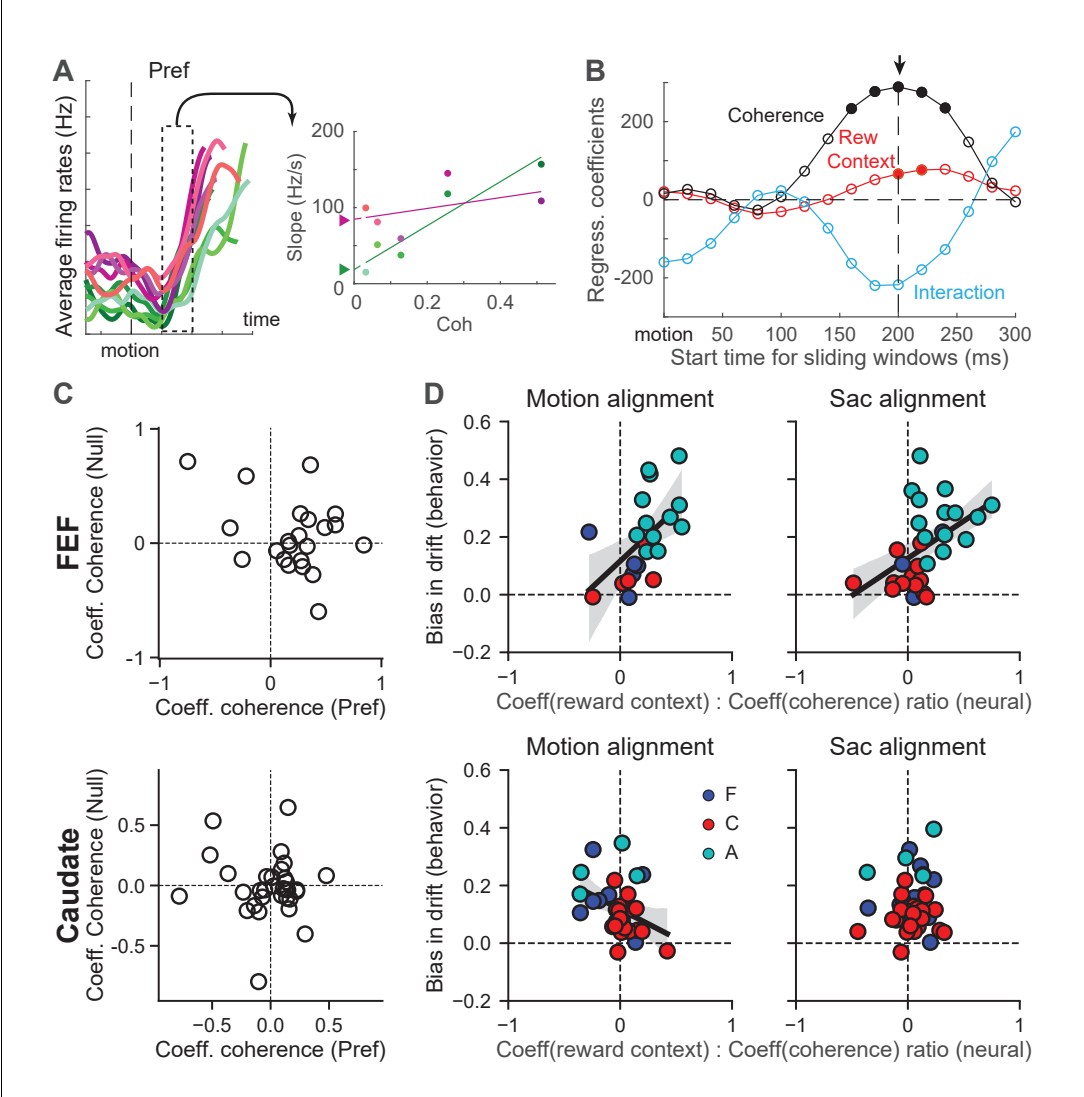

**Figure 6.** Reward-context modulation of the rate of change in FEF more closely reflected reward bias in drift rates. (A) Illustration of measurements of different modulations of the rate of change for a single neuron. Left: average firing rates of the example neuron in *Figure 2B* for its preferred choice and aligned to motion onset. In 200 ms sliding windows, linear regressions were performed to estimate the slope of firing-rate changes as a function of time, coherence, reward context and their combination. Right: slope values for the sliding window in the left panel. A multiple linear regression was performed with coherence, reward context and their interaction as the regressors (lines). The offset between the two reward contexts at zero coherence (filled triangles) represents the magnitude of reward-context modulation in the regression. (B) The regression coefficients of the linear regression for different sliding windows for the example neuron. Filled circle: coefficient was significantly different from zero (t-test, p<0.05). For each neuron, the time with the largest absolute coherence modulation was identified (arrow). For the alignment to motion onset, a minimum 100 ms visual latency was imposed. (C) Coefficient values for FEF (top) and caudate (bottom) neurons with significant coherence-modulated slope values for trials with the preferred choices. (D) Scatter plots of the ratio of regression coefficients for reward context and coherence modulation (abscissa) and the behavioral bias in drift rates (from DDM fits, ordinate), for FEF (top) and caudate (bottom) neurons with significant coherence modulation. Preferred choice only. Slope values were measured from activity aligned to motion (left) and saccade (right) onset. Line and shaded area: linear regression with significant non-zero slope (t-test, p<0.05) and 95% confidence interval. Colors indicate neurons from the three monkeys.

The online version of this article includes the following source data and figure supplement(s) for figure 6:

**Source data 1.** Source data for *Figure 6D*: FEF activity aligned to motion onset.
**Source data 2.** Source data for *Figure 6D*: FEF activity aligned to saccade onset.
**Source data 3.** Source data for *Figure 6D*: caudate activity aligned to motion onset.
**Source data 4.** Source data for *Figure 6D*: caudate activity aligned to saccade onset.
**Figure supplement 1.** Slope measurements from trials with the non-preferred choices and from neurons without consistent choice selectivity.
**Figure supplement 2.** Results of correlation analysis as used in *Figure 6D*, for the three monkeys separately.

We found that many neurons in both regions showed motion-coherence modulation of the slope of firing rates (*Figure 6*, *Table 1*), consistent with an involvement of both regions in evidence accumulation (*Ding and Gold, 2010*; *Ding and Gold, 2012b*). For choice-selective FEF neurons, the slope of change tended to be greater for higher coherence for trials with the neurons' preferred choices (i.e., positive coefficients; *Figure 6C*). Many FEF neurons also showed opposite modulation for trials with the null choices, but these effects were inconsistent, reflecting the lower reliability in estimating slope values from low firing rates. For choice-selective caudate neurons, the slope of change did not show a consistent relationship with coherence for either the preferred or null choices. The overall magnitude of the coefficients tended to be smaller for caudate neurons, reflecting the lower firing rates of caudate neurons.

FEF activity also aligned closely with the monkeys' reward-driven bias in drift rates across sessions and monkeys. All three monkeys tended to use positive reward biases in drift rates; that is toward the large-reward choice (*Figure 4E* and *Figure 6D*). The ratios of regression coefficients for reward context and coherence also tended to be positive for choice-selective neurons in the FEF (*Figure 6D*, top row). Moreover, there was a significant correlation between the monkeys' behavioral bias in drift rates and the ratio measured from neural data (Pearson's correlation coefficient: 0.55 and 0.48, p=0.0084 and 0.0077, for activity aligned to motion and saccade onset, respectively). As expected given the smaller sample sizes, none of the per-monkey results was statistically significant (*Figure 6—figure supplement 2*). These results indicated a close relationship between FEF neurons and the neural implementation of reward biases in drift rates assessed across monkeys. In the caudate sample, the behavioral bias was mostly positive, but the neural ratio was more mixed, and the two measurements did not exhibit a significant positive correlation (*Figure 6D*, bottom row; note that there was a significant negative correlation for caudate activity aligned to motion onset, correlation coefficient: −0.38, p=0.032). These results appeared inconsistent with a direct involvement of caudate neurons in implementing the reward bias in drift rates (see Discussion for a potential sampling bias).

Although the DDM does not provide predictions for neurons without consistent choice selectivity, these neurons may participate in the other aspects of decision-making, such as decision evaluation, that also uses information about the reward biases. We performed the same analysis for these neurons. In the FEF, the reward-context modulation of the slope of the firing rates also covaried with the monkeys' reward biases for activity aligned to motion onset, regardless of choices (*Figure 6—figure supplement 1C*; correlation coefficients: 0.55 and 0.38, p=0.003 and 0.018, for contralateral and ipsilateral choices, respectively). There was also a significant correlation for these not-choice-selective neurons in the caudate sample, for activity aligned to saccade onset in trials with ipsilateral choices (*Figure 6—figure supplement 1D*; correlation coefficient: 0.37, p=0.0095). Thus, FEF and caudate neurons might carry information about reward biases in drift rates for computations that are not directly related to decision formation.

In addition, a small number of neurons showed significant modulation of the slope of firing rates by the reward context-coherence interaction. In the DDM, such a modulation may relate to reward context-dependent changes in the scaling factor, *k*. However, the small sample size precluded the detection of any such relationship (data not shown).

## Reward context-modulated baseline activity was inconsistent with reward biases in relative bound heights

As we showed above, reward-driven biases in the relative bound heights of the DDM, 'bound bias' in short, can, in principle, be implemented as an offset to the beginning of the accumulation process (*Figure 4C*), an offset to the end of the accumulation process (*Figure 4D*), or the combined effects of the two. Neural activity reflecting such biases is expected to show three features. First, the neural activity should be sensitive to reward context. Second, the sign of its reward-context modulation should be congruent with the reward bias. For example, if the monkey uses the bound bias to favor the large-reward choice, when its preferred choice is paired with the large reward, then the neuron should increase its baseline firing before motion onset (as an offset to the beginning of the accumulation) or decrease its firing before saccade onset (as an offset to the end of the accumulation). Third, in consideration of our lack of knowledge of whether a neuron provides an excitatory or inhibitory role in the decision network, we can relax our expectation for sign congruency . However, we may still expect that, on trials when the reward-context modulation of neural activity is strong, the

monkey uses a larger bound bias. We tested these predictions on choice-selective neurons. Note that similar predictions cannot be specified for neurons without choice selectivity.

We found that many choice-selective neurons showed reward context-modulated activity before motion onset and/or before saccade onset in both regions (*Figure 7A–D*). We assessed the reward context modulation in running windows covering two time periods around motion onset and before saccade, respectively. *Figure 7A–D* shows heatmaps of regression coefficients for reward context using *Equation 3* (same as *Figure 3F*, but only for choice-selective neurons with significant non-zero values in any time bins). A quick glance suggested that FEF neurons tended to show positive

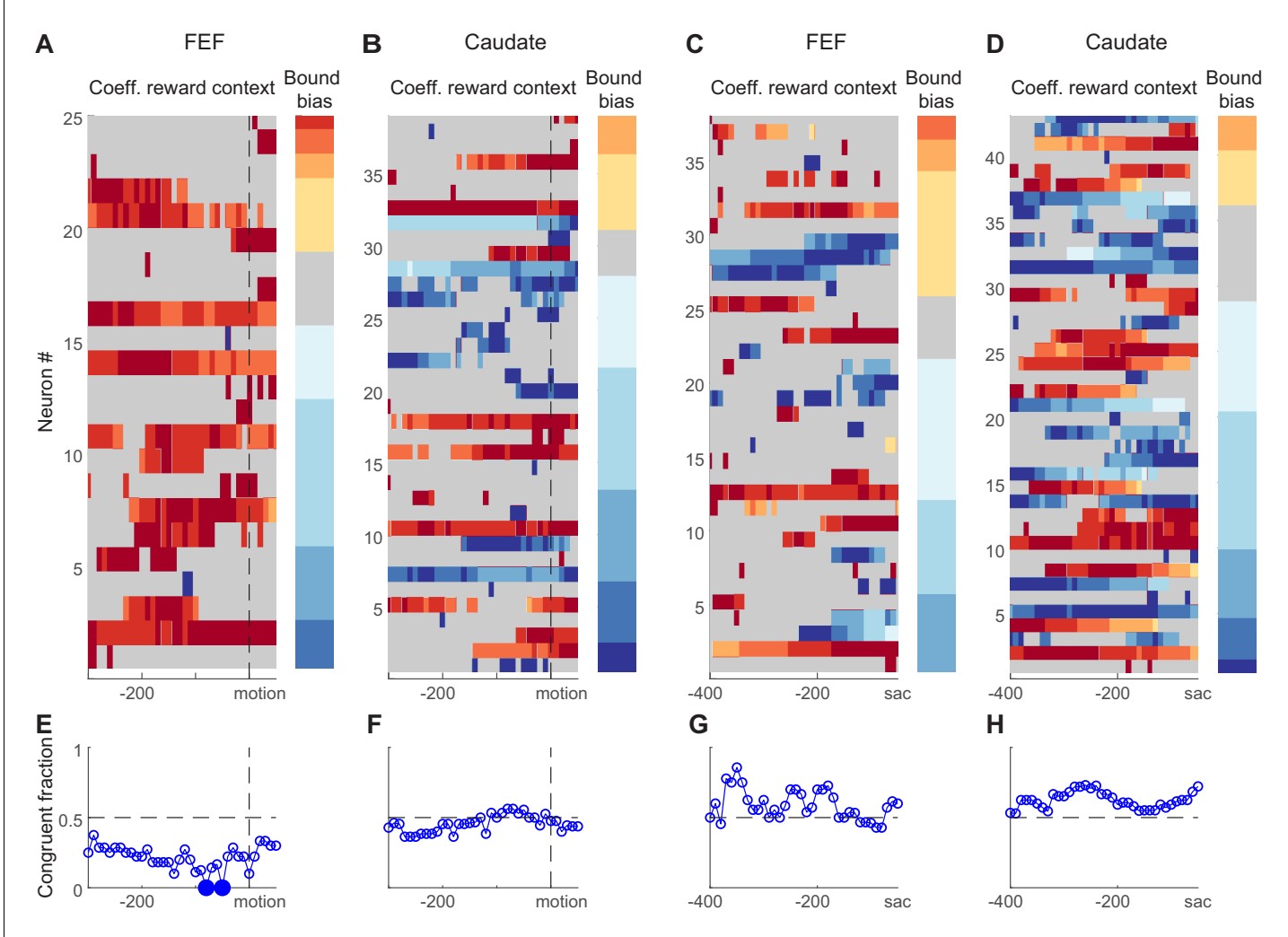

**Figure 7.** Reward-context modulation of neural activity did not conform to predictions of reward bias in relative bound heights. (A,B) Heatmaps of normalized regression coefficients for choice-selective FEF (A) and caudate (B) neural activity before/around motion onset (*Equation 3*). Only neurons with significant modulation in at least one time bin are shown (t-test, p<0.05). Neurons were sorted by bound bias values (color bar to the right), measured with DDM fits. Coefficients were normalized by the maximal absolute value for each neuron for better visualization. For the heatmaps, warm colors indicate stronger activity when the neuron's preferred choice was paired with large reward, cool colors indicate stronger activity when the null choice was paired with large reward, and gray indicates bins without significant reward context modulation. For the color bars, warm colors indicate bound biases that favored the large-reward choice, cool colors indicate bound biases that favored the small-reward choice. (C,D) Heatmaps of normalized regression coefficients for activity before saccade onset. Same format as A and B. (E-H) Fractions of neurons showing reward context modulation that was congruent with the behaviorally measured bound bias for panels (A-D), respectively. Filled circles indicate fractions that were significantly different from chance level (0.5; chi-square test, 0 < 0.05).
The online version of this article includes the following figure supplement(s) for figure 7:

**Figure supplement 1.** Reward-context modulation of neural activity did not conform to predictions of reward bias in relative bound heights.
**Figure supplement 2.** Reward-context modulation of neural activity in not-consistently-choice-selective neurons.

coefficients before motion onset (*Figure 7A*; warm colors: higher activity when the neuron's preferred choice was paired with large reward). The coefficients were more mixed in signs for FEF activity before saccade onset (*Figure 7C*) and for caudate activity (*Figure 7B and D*).

In contrast to the second expectation above, the signs of coefficients were not consistently congruent with the monkeys' behavioral bound biases. As illustrated by the color bars at the right of each panel, the monkeys tended to use negative bound biases (favoring the small-reward choice). If the neural activity reflected such biases, the heatmap should be dominated by cool colors for activity before motion onset and by warm colors for activity before saccade onset. This appeared not to be the case. We quantified the fraction of congruent sessions (*Figure 7E–H*). The only time points with fractions that differed significantly from chance suggested incongruent neural modulation for FEF activity before motion onset (Chi-square test, p=0.05, uncorrected for multiple comparisons to reduce false negatives). We also performed running regression with coherence-based regressors (*Equation 2*) and observed a similar lack of congruent modulation (*Figure 7—figure supplement 1*).

In addition to the discrepancy in signs, the magnitude of the reward-context modulation in neural activity also did not co-vary with the magnitude of the reward bias in bound heights in either region. For each neuron with significant reward-context modulation in their activity before motion onset (epoch #3 in *Figure 1*), we split the trials into two groups with larger and smaller differences in activity between reward contexts, respectively (*Figure 8A*). If the activity reflects the bound bias, we expected the former group to show a larger bound bias. We fitted the DDM to these two groups of trials and found no consistent difference in their bound biases in either brain region (*Figure 8B and C*). These results suggested that the reward context-modulated baseline activity in choice-selective FEF and caudate neurons did not directly reflect the eventual bound bias that the monkeys used.

## Discussion

Using a task with manipulations of visual stimuli and reward-choice associations, we found that a substantial fraction of neurons in both FEF and caudate were sensitive to both stimulus properties and the reward-choice association. Despite these coarse similarities, we also identified inter-regional differences in the prevalence and distribution of lateralized modulation by choice and reward context, reminiscent of previous results using tasks with either stimulus or reward-context manipulations (*Ding and Hikosaka, 2006*; *Kobayashi et al., 2007*; *Ding and Gold, 2010*; *Ding and Gold, 2012c*).

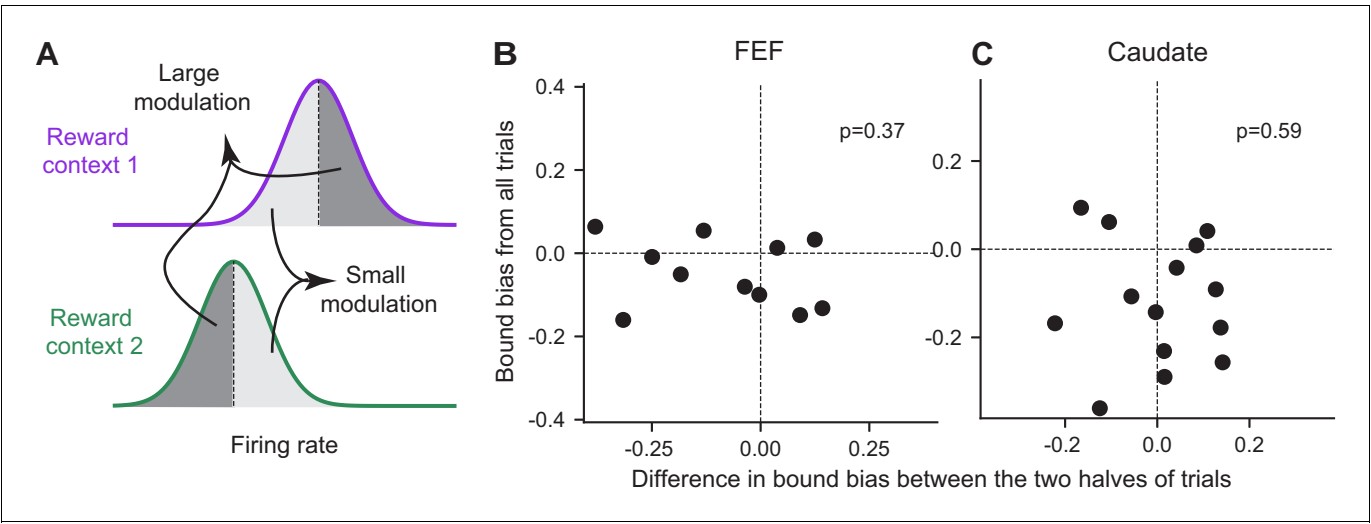

**Figure 8.** The magnitude of reward bias in relative bound heights did not vary with reward-context modulation of neural activity. (A) Trials for each reward context were split into two halves based on a neuron's average activity before motion onset (epoch #3). Reward bias in relative bound heights were measured for trials with large/small reward-context modulation of activity (dark gray/light gray). If the neural activity reflects the behavioral bias, the trials with large modulation were expected to show a larger behavioral bias. (B,C) Scatter plots of the difference in reward bias in relative bound heights between large and small-modulation trials and the bias measured from all trials for FEF (B) and caudate (C) neurons with consistent choice selectivity and significant reward context modulation. P values are from t-test (H_0: the mean difference of the x-axis values is zero).

For choice-selective FEF neurons, their average activity profile followed an accumulation-to-bound-like pattern (*Thompson et al., 1996*; *Thompson et al., 1997*; *Kim and Shadlen, 1999*; *Purcell et al., 2010*; *Ding and Gold, 2012c*). Their reward-context modulations were consistent with predictions of a reward-driven bias in drift rates, but not a reward-driven bias in relative bound heights. For choice-selective caudate neurons, their average activity profile was consistent with evidence accumulation, but not bound crossing (*Ding and Gold, 2010*). Their reward-context modulations did not show a consistent link with either form of reward-driven biases. These differences suggest that the two regions have distinct roles in implementing the computations required for this task.

The closer link between FEF activity with biases in drift rate versus bound heights may appear to be at odds with previous results implicating a more prominent role of the FEF and its rodent homolog in transforming the accumulated evidence into a categorical choice than in the evidence-accumulation process itself (*Freedman et al., 2001*; *Ferrera et al., 2009*; *Hanks et al., 2015*). However, these roles likely reflect the specific task and the subjects' strategy for performing that task. For example, for tasks involving manipulations of category definitions, FEF activity showed strong correlates of decision rules (*Freedman et al., 2001*; *Ferrera et al., 2009*). For our task, the category definitions remained constant and monkeys tended to use consistent changes in drift rates to favor the large-reward choice and variable changes in relative bound heights that can favor the large- or small-reward choice, depending on the monkey and daily session (*Fan et al., 2018*). The propensity of FEF neurons to encode the changes in drift rates may thus reflect the relative importance of those particular biases to the decision process.

FEF shares many response properties with the lateral intraparietal area (LIP), particularly for decisions based on random-dot motion stimulus (e.g., *Shadlen and Newsome, 1996*; *Kim and Shadlen, 1999*; *Roitman and Shadlen, 2002*; *Ding and Gold, 2012c*; *Meister et al., 2013*). Interestingly, a previous study of monkey LIP activity for an asymmetric-reward motion discrimination task showed opposite relationships with behavioral reward biases than what we found for FEF (*Rorie et al., 2010*): LIP activity was consistent with an involvement in reward-biased bound heights but not drift rates. The contrasts between that study and ours suggest two possible interpretations. One possibility is that LIP and FEF perform complementary roles by implementing reward biases in relative bound heights and drift rate, respectively. Another possibility is that the two regions share similar roles, and the apparent differences from the two studies reflect differences in their task designs. Rorie and colleagues used a substantially different task design from ours, including experimenter- versus subject-controlled motion viewing and trial- versus block-wise manipulations of reward contexts. In principle, these differences could influence not only what strategy monkeys use, but also which brain regions are employed to implement the required computations through training. A direct comparison between LIP and FEF neurons in the same monkeys performing the same decision task would help disambiguate these possibilities.

The lack of a consistent link between caudate activity before motion onset and reward bias in relative bound heights was surprising. Previous studies using tasks with reward manipulations and salient visual stimuli showed shared time courses between caudate activity before target onset and reward biases in RT during reward-context transitions (*Lauwereyns et al., 2002a*) and with manipulations of the timing of target onset (*Ding and Hikosaka, 2007*), as well as trial-by-trial correlations between caudate activity and action values estimated from monkeys' reward biases (*Lau and Glimcher, 2008*). We also showed previously that caudate microstimulation evoked changes in relative bound for a symmetric-reward motion discrimination task (i.e., without reward-driven biases; *Ding and Gold, 2012a*). These previous results naturally led to the hypothesis that the caudate activity preceding stimulus presentation helps to bias the starting value for evidence accumulation. Our negative results here imply that, for this task, caudate activity does not directly set the starting value. This finding is also consistent with our observation that, for the same asymmetric-reward motion discrimination task, caudate microstimulation did not cause consistent changes in bound biases (*Doi et al., 2020*). Taken together, we hypothesize that the caudate nucleus does not directly implement the bound bias, but rather coordinates the bound bias with the bias in drift rate. For simple tasks in which bound biases alone are sufficient, caudate activity appears to be directly correlated with bound bias. For complex tasks in which additional computations are involved, such a correlation could be substantially weakened.

For the lack of a relationship between caudate activity and the bias in drift rate, a caveat needs to be considered. To detect a correlation between neural activity and the reward bias in drift rates (*Figure 6*), it requires a sample that is large enough and, equally as important, with sufficient variations in behavioral biases. For practical reasons, our caudate samples were mostly from two monkeys (C and F) that tended to use smaller and less variable drift-rate biases across sessions than the other monkey (A). The more restricted ranges of drift-rate bias might have biased our results. Nevertheless, the negative correlation we observed in *Figure 6D* is consistent with our previous demonstration that caudate microstimulation tended to scale down reward biases in drift rates (*Doi et al., 2020*).

Besides decision formation, both FEF and the caudate nucleus have other hypothesized decision-related roles, including performance monitoring (*Ding and Gold, 2010*; *Ding and Gold, 2012c*; *Ding and Gold, 2013*; *Teichert et al., 2014*; *Yanike and Ferrera, 2014a*). As we showed, reward context-modulated neural activity was present in a substantial fraction of neurons that were not consistently selective for choice. These activity patterns were sensitive to reward biases in drift rates during motion viewing (*Figure 6—figure supplement 1*) or in the baseline firing before motion onset and saccade onset (*Figure 7—figure supplement 2*). In addition, some choice-selective neurons showed negative ratios of reward context and coherence coefficients (*Figure 6D*), which are not consistent with the decision variable predicted by the DDM but could reflect a choice confidence signal instead. It would be interesting to investigate further the exact functional roles of these activity patterns for solving decision-making tasks.

For our task, neither FEF nor caudate activity represented the full, latent decision variable as predicted in the DDM framework. For example, in addition to the disconnect between bound bias and reward context-modulated baseline activity, the example FEF neuron in *Figure 6A* showed a strong modulation by the coherence-reward context interaction, which was not predicted by the DDM. A striking observation for FEF was the relatively consistently opposite signs in the reward bias in bound heights and the reward-context modulation of pre-motion baseline activity in choice-selective neurons. This finding raises several possibilities, including: (1) the DDM framework does not accurately capture the monkeys' decision-related computations; (2) the reward-context modulation of pre-motion baseline activity contributes to the reward bias in bound heights through an intermediary, sign-reversing mechanism; and/or (3) such activity does not contribute to the reward bias in bound heights. Relevant to the first possibility, we previously fitted the monkeys' performance using two model variants (fixed-bound and collapsing-bound) and two fitting procedures (Hierarchical DDM using MCMC sampling and single-session DDM fits with multiple runs using maximum a posteriori) (*Fan et al., 2018*; *Doi et al., 2020*). These different ways of model fitting resulted in similar patterns of the signs of reward biases in bound heights and drift rates. These data argued against gross inaccuracy in DDM fits of reward bias in bound heights, but it remains to be tested whether a non-DDM framework could capture the monkey's performance and predict modulations of decision variables more in line with those observed in FEF activity.

Many other brain regions undoubtedly contribute to decisions that combine sensory and reward information. These regions likely include the lateral intraparietal area (LIP), superior colliculus, and the premotor cortex in monkeys, each of which has been shown to represent the basic patterns of activity that are reminiscent of an accumulation-to-bound decision process (*Roitman and Shadlen, 2002*; *Ratcliff et al., 2003*; *Thura and Cisek, 2016*). Even more regions have shown reward-context modulated pre-stimulus activity and saccade-related activity in monkeys performing tasks with reward manipulations and salient visual stimuli, including those areas and many other nuclei in the basal ganglia (*Platt and Glimcher, 1999*; *Coe et al., 2002*; *Lauwereyns et al., 2002a*; *Sato and Hikosaka, 2002*; *Ikeda and Hikosaka, 2003*; *Roesch and Olson, 2003*; *Isoda and Hikosaka, 2008*). Reward manipulations also likely affect the sensory representation itself, leading to biased drift rates (*Cicmil et al., 2015*). More studies like ours that directly compare neural activity in different brain regions under the same task conditions are needed to better understand their overlapping and distinct roles in these kinds of decisions. A particularly intriguing target of such studies would be the superior colliculus, because of its convergent inputs from FEF, LIP, and the basal ganglia, as well as its well-documented roles in attentional control that are likely closely related to reward modulation (*Krauzlis et al., 2018*). The distribution of neural representations of biases in drift rates and relative bound heights would also help us understand the dissociated effects of Parkinson's Disease on these two forms of bias (*Perugini et al., 2016*).

To summarize, FEF and caudate activity showed modulations by choice, reward context, and visual stimulus strength, in monkeys that combined reward context and visual input into categorical saccade choices. These two regions shared certain features in their activity, but also showed distinct patterns that implicated their different roles in complex decision making. It would be interesting to examine how the relative contributions of FEF and caudate neurons develop over training (*Antzoulatos and Miller, 2011*; *Seo et al., 2012*) and with induced changes in the subjects' reward bias strategy.

# Materials and methods

**Key resources table**

| Reagent type (species) or resource | Designation | Source or reference | Identifiers | Additional information |
|---|---|---|---|---|
| Software, algorithm | MATLAB | Mathworks | RRID:SCR_001622 | https://www.mathworks.com |
| Software, algorithm | Python 3.5 | Python Software Foundation | RRID:SCR_008394 | https://www.python.org/ |
| Software, algorithm | Psychophysics Toolbox | *Pelli, 1997*; *Kleiner, 2007* | RRID:SCR_002881 | http://psychtoolbox.org/ |
| Software, algorithm | Pandas v0.19.2 | Python Data Analysis Library | RRID:SCR_018214 | https://pandas.pydata.org/ |
| Software, algorithm | Scikit-learn v0.18.1 | scikit-learn.org | RRID:SCR_002577 | https://scikit-learn.org/stable/ |
| Software, algorithm | Statsmodels v0.8.0 | Statsmodels.org | RRID:SCR_016074 | https://www.statsmodels.org/stable/index.html |
| Software, algorithm | Scipy v0.18.1 | SciPy.org | RRID:SCR_008058 | https://docs.scipy.org/doc/scipy/reference/stats.html |
| Software, algorithm | PyMC 2.3.6 | http://github.com/pymc-devs/pymc | | http://github.com/pymc-devs/pymc |
| Software, algorithm | dPCA | *Kobak et al., 2016* | | https://github.com/machenslab/dPCA/tree/master/matlab |

## Experimental model and subject details

All training and experimental procedures were in accordance with the National Institutes of Health Guide for the Care and Use of Laboratory Animals and were approved by the University of Pennsylvania Institutional Animal Care and Use Committee (protocol #804726). Details about monkey training, behavioral tasks, and caudate recording were reported previously (*Fan et al., 2018*; *Doi et al., 2020*).

## Neural recording

Each monkey was implanted with a head holder and recording cylinder that provided access to the FEF (right for monkeys C and F, left for monkey A). The FEF was identified as the anterior bank of the arcuate sulcus where saccades were evoked with microstimulation of <50 µA (70 ms trains of 300 Hz, 250-µs biphasic pulses) (*Bruce and Goldberg, 1985*; *Ding and Gold, 2012b*). Neural activity was recorded using a combination of glass-coated tungsten electrodes (Alpha-Omega), epoxylite-coated tungsten electrodes (FHC), and multi-contact electrodes (V-probe, Plexon, Inc; Multitrodes, Thomas Recording), driven by a NaN microdrive (NAN Instruments, LTD). A memory-guided delayed saccade task was used to estimate the response field of a neuron (*Ding and Gold, 2012b*). For the motion discrimination task, one choice target was placed in the response field and the other was placed symmetrically across the central fixation point. Motion directions were along the axis defined by the choice targets.

Single-unit recordings were obtained for neurons that showed activity modulation during trials by visual inspection and single-unit spikes were sorted offline (OfflineSorter, Plexon). Neurons with low firing rates (peak firing rate <5 Hz) and few trials (<5 finished trials per choice × coherence × reward context combination or <3 correct trials per combination) were excluded from analysis.

## Behavioral analysis

To quantify reward context-induced biases, a logistic function was fitted to the choice data for all trials for each session:

$$P_{contra\ choice} = \frac{1}{1 + e^{-Slope(Coh+Bias)}},$$ (1)

where *Coh* is the signed motion coherence,

$$Slope = slope_0 + slope_{rew} \times RewCont,$$

$$Bias = bias_0 + bias_{rew} \times RewCont,$$

$$RewCont = \{1\ for\ contralateral - large\ reward\ blocks, -1\ for\ ipsilateral - large\ reward\ blocks\}.$$

To infer the latent decision variable, we also fitted the choice and saccade reaction time (RT) data simultaneously to a drift-diffusion model (DDM; *Figure 4A*), following previously established procedures (*Fan et al., 2018*). We defined RT as the time from stimulus onset to saccade onset. Saccade onset was identified offline with respect to velocity (>40°/s) and acceleration (>8000°/s²). The DDM assumes that the latent decision variable (DV) is the time integral of evidence (*E*) and reward asymmetry-induced fictive evidence (*me*), scaled by a constant (*k*).

$$E \sim N(coherence, 1)\ and\ DV = \int k(E + me)\ dt$$

At each time point, the DV was compared with two collapsing choice bounds (*Zylberberg et al., 2016*). The time course of the choice bounds was specified as $a/\left(1 + e^{\beta\_alpha(t-\beta\_d)}\right)$, where $\beta\_alpha$ and $\beta\_d$ controlled the rate and onset of decay, respectively and $a$ specified the maximal distance between the two choice bounds. A bias-related parameter (*z*) specified the relative bound heights of the two choice bounds, where z = 0.5 indicated equal bound heights for the two choices, z>0.5 indicated that the upper bound was closer to the starting point of evidence accumulation than the lower bound.

For sessions with neurons showing choice-selective activity during a pre-saccade period (see below for epoch definitions), the upper bound was associated with the preferred choice and the lower bound was associated with the null choice. In other words, if DV crossed the upper bound first, a saccade was made to the target inside the neuron's response field; if DV crossed the lower bound first, a saccade was made to the other target.

DDM model fitting was performed, separately for each session, using the maximum a posteriori estimate method (python v3.5.1, pymc 2.3.6) and prior distributions suitable for human and monkey subjects (*Wiecki et al., 2013*). We performed at least five runs for each variant and used the run with the highest likelihood for further analyses. Biases in drift and bound (*Figure 5I*) were computed as the difference in the fitted *me* and *z* values between the two reward contexts, respectively. Positive values indicated biases toward the large reward choice.

## Neural data analysis

We performed three regression analyses on the neural data. First, for each single unit, we computed the average firing rates in eight task epochs (*Figure 1A*): three epochs before motion stimulus onset (400 ms window beginning at target onset, variable window from target onset to dots onset, and 400 ms window ending at motion onset), two epochs during motion viewing (a fixed window from 100 ms after motion onset to 100 ms before median RT and a variable window from 100 ms after motion onset to 100 ms before saccade onset), a pre-saccade 100 ms window, a peri-saccade 300 ms window beginning at 100 ms before saccade onset, and a post-saccade 400 ms window beginning at saccade onset (before feedback and reward delivery). For each unit, a multiple linear regression was performed on the spike counts in correct trials, for each task epoch separately.

$$Spike\ count = \beta_0 + \beta_{Choice} \times I_{Choice} + \beta_{RewCont} \times I_{RewCont} + \beta_{RewSize} \times I_{RewSize}$$
$$+ \beta_{Coh-Contra} \times I_{Coh-Contra} + \beta_{Coh-Ipsi} \times I_{Coh-Ipsi} \qquad (2)$$
$$+ \beta_{RewCoh-Contra} \times I_{Coh-Contra} \times I_{RewSize} + \beta_{RewCoh-Ipsi} \times I_{Coh-Ipsi} \times I_{RewSize},$$

where

$$I_{Choice} = \{1\ \text{for contralateral/up choice},\ -1\ \text{for ipsilateral/down choice}\},$$

$$I_{RewCont} = \{1\ \text{for contralateral/up} - \text{large reward blocks}, -1\ \text{for ipsilateral/down} - \text{large reward blocks}\},$$

$$I_{RewSize} = \{1\ \text{if a large reward is expected for the choice},\ -1\ \text{if a small reward is expected}\},$$

$$I_{Coh-Contra} = \{\text{coherence for contralateral/up choice},\ 0\ \text{for ipsilateral/down choice}\},$$

and

$$I_{Coh-Ipsi} = \{0\ \text{for contralateral/up choice, coherence for ipsilateral/down choice}\}.$$

Significance of non-zero coefficients was assessed using *t*-test (criterion: p=0.05).

Second, for each single unit, we also performed running regressions using *Equation 2* on the spike counts within 150 ms windows every 10 ms. These running regressions were performed on activity aligned to target, motion, and saccade onsets separately. Only correct trials were included. Time windows with fewer than 10 correct trials were excluded.

Third, for these neurons, the following multiple linear regressions was performed in epochs and running windows defined above:

$$Spike\ count = \beta_0 + \beta_{Choice} \times I_{Choice} + \beta_{RewCont} \times I_{RewCont} + \beta_{RewSize} \times I_{RewSize}$$
$$+ \beta_{RT-Contra} \times RT_{Contra} + \beta_{RT-Ipsi} \times RT_{Ipsi} \qquad (3)$$
$$+ \beta_{RewRT-Contra} \times RT_{Contra} \times I_{RewSize} + \beta_{RewRT-Ipsi} \times RT_{Ipsi} \times I_{RewSize}$$

where

$$RT_{Contra} = \{\text{RT for the contralaeteral/up choice},\ 0\ \text{for the ipsilateral/down choice}\},$$

and

$$RT_{Ipsi} = \{0\ \text{for the ipsilateral/down choice},\ \text{RT for the contralateral/up choice}\}.$$

To control for reward context or choice-dependent modulation of RT, the RT values used in the regressions were the mean-subtracted values, with the mean values measured for the corresponding reward context-choice combinations. Significance of non-zero coefficients was assessed using *t*-test (criterion: p=0.05).

Because reward context was alternated in blocks in our task, we examined whether the significant coefficients for reward context in these regressions were simply due to serial correlation of the reward context values (*Elber-Dorozko and Loewenstein, 2018*). To assess the potential effect of serial correlation on our results, we focused on the epoch-based regressions. For each neuron x epoch combination with a significant coefficient for reward context, we estimated the null distribution of the coefficient by performing 100 regressions using random, unmatched reward-context values. To obtain these unmatched values, we concatenated the reward-context values from all neurons and randomly picked a segment for each regression. We performed one-tailed comparisons between the null distributions and the coefficients obtained using real data and updated the p-values for the reward-context coefficient accordingly. To gain another perspective of the task-related modulation patterns in each population, we performed demixed principal component analysis on spike activity (*Kobak et al., 2016*), using the publicly available source code (https://github.com/machenslab/dPCA/tree/master/matlab). We focused on two epochs for activity aligned to motion and saccade onset, respectively (*Figure 3—figure supplements 4–7*) and used only correct trials for this analysis. To mitigate the unbalance inherent in our data set (e.g., there were fewer correct trials

for low coherence or when the choice led to small reward; different coherence levels were used for the three monkeys), we used the four highest coherence levels for each session as equivalent conditions across monkeys/sessions.

## Measuring the slope of change in firing rates

Only correct trials were included for this analysis. Spike trains were aligned to motion onset and grouped by coherence x reward context combinations. The average firing rates were computed for each combination, truncated at median RT for the combination, and convolved with a Gaussian kernel ($\sigma$ = 20 ms). The slope of change was measured from 200 ms running windows (in 20 ms steps) of the smoothed firing rates for each combination, using a linear regression with time as the independent variable. For each running window, a multiple linear regression was performed, using coherence, reward context, and their interaction as the independent variable and the slopes of change as the dependent variable. Significance for individual regressors was assessed using $t$-test (criterion: p=0.05).

## Splitting trials based on baseline activity before motion onset

This analysis was performed on neurons with significant reward-context modulation of average firing rates during epoch #3 (a 400 ms window before motion onset), as identified using the regression in *Equation 3*. For each neuron, trials were divided into two halves based on the average firing rate in epoch #3, separately for each reward context. This resulted in four combinations of trials: high/low firing rates and two reward contexts (*Figure 8A*). The 'large modulation' trials comprised of high-firing-rates trials in the neuron's preferred reward context and low-firing-rates trials in the other context. Conversely, the 'small modulation' trials comprised of low-firing-rates trials in the neuron's preferred reward context and high-firing-rates trials in the other context. If the neural activity is closely linked to the reward bias in relative bound heights, the trials with large modulation were expected to show a larger reward bias. These two types of trials were fitted by DDM separately and their estimated reward bias in relative bound heights were compared (*Figure 8B and C*).

## Acknowledgements

We thank Jean Zweigle for animal care and Drs. Kae Nakamura and Takahiro Doi for helpful comments. This work was supported by NIH National Eye Institute (R01-EY022411; LD and JIG), University of Pennsylvania (University Research Foundation Pilot Award; LD), and Hearst Foundations Graduate student fellowship (YF).

## Additional information

### Competing interests

Joshua I Gold: Senior Editor, eLife. The other authors declare that no competing interests exist.

### Funding

| Funder | Grant reference number | Author |
|---|---|---|
| National Institutes of Health | R01-EY022411 | Long Ding Joshua I Gold |
| University of Pennsylvania | University Research Foundation Pilot Award | Long Ding |
| Hearst Foundations | Graduate student fellowship | Yunshu Fan |

The funders had no role in study design, data collection and interpretation, or the decision to submit the work for publication.

### Author contributions

Yunshu Fan, Data curation, Formal analysis, Investigation, Methodology, Writing - review and editing; Joshua I Gold, Conceptualization, Funding acquisition, Visualization, Writing - review and

editing; Long Ding, Conceptualization, Resources, Data curation, Software, Formal analysis, Supervision, Funding acquisition, Validation, Investigation, Visualization, Methodology, Writing - original draft, Project administration, Writing - review and editing

### Author ORCIDs
Yunshu Fan https://orcid.org/0000-0003-2597-5173
Joshua I Gold https://orcid.org/0000-0002-6018-0483
Long Ding https://orcid.org/0000-0002-1716-3848

### Ethics
Animal experimentation: All training and experimental procedures were in accordance with the National Institutes of Health Guide for the Care and Use of Laboratory Animals and were approved by the University of Pennsylvania Institutional Animal Care and Use Committee (protocol #804726). Details about monkey training, behavioral tasks, and caudate recording were reported previously (Fan et al., 2018; Doi et al., 2020).

### Decision letter and Author response
Decision letter https://doi.org/10.7554/eLife.60535.sa1
Author response https://doi.org/10.7554/eLife.60535.sa2

## Additional files

### Supplementary files
• Transparent reporting form

### Data availability
Data used for this manuscript are included as supporting files.

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
