## [Decision Letter]

Thank you for submitting your article "Frontal eye field and caudate neurons encode complementary features of reward-biased perceptual decisions" for consideration by *eLife*. Your article has been reviewed by three peer reviewers, one of whom is a member of our Board of Reviewing Editors, and the evaluation has been overseen by Michael Frank as the Senior Editor. The following individuals involved in review of your submission have agreed to reveal their identity: Chandramouli Chandrasekaran (Reviewer #2); Jochen Ditterich (Reviewer #3).

The reviewers have discussed the reviews with one another and the Reviewing Editor has drafted this decision to help you prepare a revised submission.

Summary:

This study examined how the activity of neurons in the two important nodes in the frontocortico-ganglia network, namely, the frontal eye field (FEF) and caudate nucleus (CD), change their activity according to the animal's choice, incoming sensory information, and expected outcome, during a asymmetrically rewarded random-dot motion discrimination task. The main focus of this manuscript is to report significant differences in the properties of neurons in these two structures, and how they might covary with the systematic changes in the parameters of the drift-diffusion model (DDM). The main differences between the FEF and CD were (1) that choice signals were more robust in the FEF, (2) that reward context signals were more robust in the CD, and (3) that the coherence signals lasted longer after the choice in the CD. These results were obtained using the conventional regression analyses and appear to be reliable. By contrast, the analyses that attempted to link the changes in the FEF and CD activity to the behavioral variability across different sessions were confusing and not convincing, and require significant additional work.

Essential revisions:

1) All the reviewers felt that the reward-dependent changes in the baseline firing rate before motion set must be analyzed more thoroughly and discussed better, especially given the opposite conclusion in a previous report by Rorie and Newsome ("Overall, detailed analysis and computer simulation reveal that our data are consistent with a two-stage drift diffusion model proposed by Diederich and Bussmeyer, 2006 for the effect of payoffs in the context of sensory discrimination tasks. Initial processing of payoff information strongly influences the starting point for the accumulation of sensory evidence, while exerting little if any effect on the rate of accumulation of sensory evidence.").

The relationship between the result in Figure 3F, middle column, suggests that, at the time of motion onset, a substantial fraction of FEF neurons are modulated by reward context, while the result in Figure 7B show that this modulation was not particularly congruent with relative bound heights in the DDM. This might be possible when there is a sizeable number of sessions with a positive decision bound asymmetry and a sizeable number of sessions with a negative decision bound asymmetry (which seems consistent with what is plotted in Figure 4I), although the sign of the behavioral bias does not change. However, it is very surprising to see Figure 7D, which shows hardly any sessions with a positive bias in bound heights. This should be explained better.

Related to this, when assessing congruency for the statistical analysis shown in Figure 7B, did the authors use only the sessions with context modulations and biases that were significantly different from zero (i.e., the red dots in Figure 7—figure supplement 2)? Or all sessions (i.e., you also used the sign of the parameters to assess congruency, even when the parameters were not both significantly different from zero, meaning you used all of the dots in Figure 7—figure supplement 2, regardless of their color)? If the latter, do they get the same result when limiting your analysis to the red dots? It is possible that the noise contributed by parameters with very small absolute values could have differentially affected the results for the different brain areas, as the dissociation is based on the fraction of congruent modulations being significantly different from 0.5 in caudate neurons, but not so for FEF neurons.

2) The reviewers are concerned about the fact that the caudate data were collected from monkeys C and F, whereas the majority of FEF data were collected from a different animal, monkey A. It is therefore important to make sure that the reported dissociation is indeed one between different brain areas, and not one between different monkeys. Figure 1B, middle row and Figure 4I suggest that the different monkeys were using somewhat different strategies when biasing their decisions, and such differences could be reflected in the neural data. We hope that this concern can be addressed with the existing dataset, but collecting additional data (caudate data from monkey A or more FEF data from monkeys C and/or F) would also be an option. Can the same dissociation be demonstrated when restricting the analysis to data from monkeys C and F? If not, the authors should come up with alternative strategies for demonstrating that the results are not related to monkey identity.

3) The heterogeneity across different neurons might be better handled by a more modern method, such as Targeted Dimensionality Reduction or demixed PCA, which would provide more rigorous and easier interpretability. For example, Gaussian-smoothed firing rates can be used as input to dPCA and the target + motion epochs and another one aligned to movement onset. This would allow readers who think more in terms of neural populations to appreciate this paper more. Interaction terms etc can be easily included in the analysis.

4) Alternative hypotheses, such as two stage model, pre-stimulus urgency signals, starting point hypothesis, need to be tested and rejected more convincingly. Currently, much larger effects in baseline state across a large neural population in FEF is deemphasized compared to the smaller effects seen in the non-choice selective caudate neurons. This might be because the analysis of neural data is based on the conviction that their behavioral model is also the best model for the neural data, but it might be necessary to explore the possibilities beyond their best behavioral model, which might still be wrong. This is also where population analysis (e.g., dPCA) might be helpful. On the one hand, it may be reasonable to only select choice-selective neurons but in doing so we are tossing 70% of the dataset. Only 44 and 36 neurons are now going into your analysis. If the authors decode variables from dPCA with reasonable variance you can just directly look at them in relation to model predictions. It uses the whole dataset and that is the advantage of such an approach.

5) The regression analysis used in this analysis shown in Figure 6 was based on the rate of change in firing rate (not the firing rate itself), but how the slope was calculated was not explicitly explained. The details of this should be given in the Materials and methods. For example, to cleanly separate the effect related to drift rate, it might be necessary to remove the contribution from the changes in the baseline firing rate, and simply calculating the slope of the regression model applied to different bins of the neural activity during the 200 ms window might not be sufficient for this.

Also, isn't it possible that at least some of the effects illustrated in Figure 6C are mediated by the changes in neural activity related to RT? If so and if the contribution of RT is not controlled for, how does this affect the interpretation?

6) To distinguish between different scenarios depicted in Figure 4, the authors applied a regression model that relates to the slope of firing rate to coherence, reward context, and their interaction. However, since this model includes the interaction term, the ratio between the regression coefficients for the two main terms is not the most appropriate quantity to test the scenario in Figure 4B. An alternative and simpler method might be to examine the coefficient for coherence and the difference in the average slope in the two reward contexts. For example, although the example neuron in Figure 6A might have a significant effect of reward context (in terms of intercept), the effect of reward context varies with coherence effect and reverses for high coherence, which is not consistent with the pattern expected from the biased drift model (Figure 4B). In addition, the ratio between the two regression coefficients might produce unreliable results, because they are disproportionally influenced by the denominator (log transformation might be appropriate). The negative ratios shown for some neurons are also difficult to interpret.

7) The dissociation between Figure 6C and D is based on finding a significant correlation in C, but not in D. The result would be strengthened by being able to show that there is a statistical difference between one particular parameter that can be estimated for both FEF and caudate neurons. For example, is it possible to estimate both linear regression slopes and to show that they are statistically different?

8) To test whether reward-related changes in neural activity was related to the variability in the bound height, they focused on the consistency in the signs of the regression coefficients for reward context and bias in the bound heights (Figure 7D). However, these coefficients are negative for the majority of the neurons, so they do not address the question of whether the variability in these two measures are correlated across sessions (for example, what was the correlation coefficient for the data shown in Figure 7D?).

9) The authors have used two regression models to examine the activity of FEF and CD neurons, one including coherence (sensory variable), and the other including RT (motor variable). Since the effects of these variables were modeled separately for each choice (e.g., ipsi vs. contra), they are correlated with each other, but in principle, this should not prevent the authors from including both of these regressors in the same model. This would be preferred, because it is likely that these two factors still influence the activity in either or both of these structures differently. Also, when the effects of coherence and RT were summarized in Figure 3, it would be better to correct them for multiple comparisons. In addition, the reward context was varied across blocks, which would introduce serial correlation in both independent and dependent variables of the regression model, making a standard t-test no longer appropriate (c.f., Elber-Dorozko and Loewenstein, 2018).

[Editors' note: further revisions were suggested prior to acceptance, as described below.]

Thank you for resubmitting your work entitled "Frontal eye field and caudate neurons encode complementary features of reward-biased perceptual decisions" for further consideration by *eLife*. Your revised article has been evaluated by Michael Frank (Senior Editor) and a Reviewing Editor.

The manuscript has been improved but there are some remaining issues that need to be addressed before acceptance, as outlined below:

This is a significant study that advances our understanding on the functional specialization of the frontal eye field and caudate nucleus during a perceptual decision making with asymmetric reward outcomes. The reviewers appreciated the extensive amount of work performed by the authors to address their previous concerns, but have one important remaining concern. In particular, the authors are concerned that the relationship between the behavioral bias in drift and the neural effects of reward context seen in the FEF (Figure 6) might be driven by the results from one animal (monkey A). In addition, it is possible that the results might be due to the individual differences among different monkeys rather than the session-by-session (or neuron-by-neuron) changes in behaviors (i.e., Simpson's paradox). Therefore, it is strongly suggested that the authors should analyze and report the results separately for individual animals, or at least repeat this analysis after excluding the results from monkey A.

---

## [Author Response]

Essential revisions:1) All the reviewers felt that the reward-dependent changes in the baseline firing rate before motion set must be analyzed more thoroughly and discussed better, especially given the opposite conclusion in a previous report by Rorie and Newsome ("Overall, detailed analysis and computer simulation reveal that our data are consistent with a two-stage drift diffusion model proposed by Diederich and Bussmeyer, 2006 for the effect of payoffs in the context of sensory discrimination tasks. Initial processing of payoff information strongly influences the starting point for the accumulation of sensory evidence, while exerting little if any effect on the rate of accumulation of sensory evidence.").

We thank the reviewers for this comment. We understand that this negative finding is contrary to what the field, including ourselves, have hypothesized for these two brain regions and thus requires more thorough documentation. We therefore completely revised the section about the baseline activity the presentation of results regarding the baseline activity, including new figures and analysis results to illustrate the lack of expected relationships between reward-context modulation of baseline activity and behavioral bias in bound heights, in terms of either the sign or magnitude of reward modulation. We also emphasized the difference between our FEF data and Rorie and colleagues’ LIP data in the Discussion:

“FEF shares many response properties with the lateral parietal area (LIP), particularly for decisions based on random-dot motion stimulus (e.g., Shadlen and Newsome, 1996; Kim and Shadlen, 1999; Roitman and Shadlen, 2002; Ding and Gold, 2012c; Meister et al., 2013). Interestingly, a previous study of monkey LIP activity for an asymmetric-reward motion discrimination task showed opposite relationships with behavioral reward biases than what we found for FEF (Rorie et al., 2010): LIP activity was consistent with an involvement in reward-biased bound heights but not drift rates. The contrasts between that study and ours suggest two possible interpretations. One possibility is that LIP and FEF perform complementary roles by implementing reward biases in relative bound heights and drift rate, respectively. Another possibility is that the two regions share similar roles, and the apparent differences from the two studies reflect differences in their task designs. Rorie and colleagues used a substantially different task design from ours, including experimenter- versus subject-controlled motion viewing and trial- versus block-wise manipulations of reward contexts. In principle, these differences could influence not only what strategy monkeys use, but also which brain regions are employed to implement the required computations through training. A direct comparison between LIP and FEF neurons in the same monkeys performing the same decision task would help disambiguate these possibilities.”

The relationship between the result in Figure 3F, middle column, suggests that, at the time of motion onset, a substantial fraction of FEF neurons are modulated by reward context, while the result in Figure 7B show that this modulation was not particularly congruent with relative bound heights in the DDM. This might be possible when there is a sizeable number of sessions with a positive decision bound asymmetry and a sizeable number of sessions with a negative decision bound asymmetry (which seems consistent with what is plotted in Figure 4I), although the sign of the behavioral bias does not change. However, it is very surprising to see Figure 7D, which shows hardly any sessions with a positive bias in bound heights. This should be explained better.

We shared with the reviewers the same prior expectation that reward context-modulated baseline activity in FEF neurons should be directly linked to behavioral bound asymmetry, i.e., the signs should be congruent. We were surprised to find that this was not the case. We have shown and discussed extensively in Fan et al., 2018 that the monkeys tended to use negative bound biases (i.e., in the non-adaptive direction, to compensate for excessive drift-rate biases). We made new figures to show that, despite the dominance of behavioral bound bias favoring the small-reward choice, FEF neurons tended to show higher baseline activity when their preferred choice was paired with large reward (Figure 7A) and caudate neurons showed a roughly even mix of higher or lower activity for this reward context. We performed additional analysis using trials split into two groups, with larger and smaller reward-context modulation of baseline activity, respectively. Contrary to the prediction that the behavioral reward bias should be larger for the former group, we did not find any consistent differences between the groups (Figure 8).

We added our interpretation of these results in Discussion:

“A striking observation for FEF was the relatively consistently opposite signs in the reward bias in bound heights and the reward-context modulation of pre-motion baseline activity in choiceselective neurons. This finding raises several possibilities, including: 1) the DDM framework does not accurately capture the monkeys’ decision-related computations; 2) the reward-context modulation of pre-motion baseline activity contributes to the reward bias in bound heights through an intermediary, sign-reversing mechanism; and/or 3) such activity does not contribute to the reward bias in bound heights. Relevant to the first possibility, we previously fitted the monkeys’ performance using two model variants (fixed-bound and collapsing-bound) and two fitting procedures (Hierarchical DDM using MCMC sampling and single-session DDM fits with multiple runs using maximum a posteriori) (Fan et al., 2018; Doi et al., 2020). These different ways of model fitting resulted in similar patterns of the signs of reward biases in bound heights and drift rates. These data argued against gross inaccuracy in DDM fits of reward bias in bound heights, but it remains to be tested whether a non-DDM framework could capture the monkey’s performance and predict modulations of decision variables more in line with those observed in FEF activity.”

Related to this, when assessing congruency for the statistical analysis shown in Figure 7B, did the authors use only the sessions with context modulations and biases that were significantly different from zero (i.e., the red dots in Figure 7—figure supplement 2)? Or all sessions (i.e., you also used the sign of the parameters to assess congruency, even when the parameters were not both significantly different from zero, meaning you used all of the dots in Figure 7—figure supplement 2, regardless of their color)? If the latter, do they get the same result when limiting your analysis to the red dots? It is possible that the noise contributed by parameters with very small absolute values could have differentially affected the results for the different brain areas, as the dissociation is based on the fraction of congruent modulations being significantly different from 0.5 in caudate neurons, but not so for FEF neurons.

In the original manuscript, the results in Figure 7B used only sessions with significant reward context modulation (i.e., the red dots in Figure 7—figure supplement 2). The fraction of congruent modulation was thus not an artifact from very small absolute values. Note that we have made major revisions in Figure 7 and those original figures were removed.

2) The reviewers are concerned about the fact that the caudate data were collected from monkeys C and F, whereas the majority of FEF data were collected from a different animal, monkey A. It is therefore important to make sure that the reported dissociation is indeed one between different brain areas, and not one between different monkeys. Figure 1B, middle row and Figure 4I suggest that the different monkeys were using somewhat different strategies when biasing their decisions, and such differences could be reflected in the neural data. We hope that this concern can be addressed with the existing dataset, but collecting additional data (caudate data from monkey A or more FEF data from monkeys C and/or F) would also be an option. Can the same dissociation be demonstrated when restricting the analysis to data from monkeys C and F? If not, the authors should come up with alternative strategies for demonstrating that the results are not related to monkey identity.

The reviewers raised an important caveat for our interpretation. Given the noise in both behavioral and neural measurements, a sizeable range of drift-rate bias values is necessary to detect any relationship between the behavioral drift-rate bias and its neural representation. As we showed before (Fan et al., 2018) and here (Figure 4E), monkey A tended to use larger driftrate biases than monkeys C and F, while the latter two monkeys tended to use smaller and, more importantly for this analysis, less variable drift-rate biases across sessions. The differences in the daily variations among monkeys in the FEF and caudate samples may contribute to the apparent inter-regional differences.

We have obtained caudate recordings from monkey A (before it was euthanized for clinical reasons) and applied the same inclusion criteria for all caudate and FEF recordings, resulting in 18, 49, and 73 caudate neurons and 85, 24, and 40 FEF neurons from monkeys A, C and F, respectively. There was a significant positive cross-neuron/session correlation (conforming to DDM predictions) between reward context-modulation of the slope of FEF, but not caudate, activity and reward bias in drift rate (Figure 6). The inclusion of monkey A’s data actually resulted in a significant negative correlation for the caudate sample. That is, the sampling of different monkeys changed the correlation, but the difference between FEF and caudate neurons holds.

As mentioned above, for analysis related to bound bias, we removed the results from not consistently choice-selective neurons in both regions. We added new figures and analyses, but the basic finding holds for choice-selective neurons: reward-context modulation of baseline activity in both FEF and caudate do not appear to be closely linked to reward biases in relative bound heights.

We completely revised the text in the last two sections of Results accordingly.

3) The heterogeneity across different neurons might be better handled by a more modern method, such as Targeted Dimensionality Reduction or demixed PCA, which would provide more rigorous and easier interpretability. For example, Gaussian-smoothed firing rates can be used as input to dPCA and the target + motion epochs and another one aligned to movement onset. This would allow readers who think more in terms of neural populations to appreciate this paper more. Interaction terms etc can be easily included in the analysis.

We performed the dPCA as the reviewers suggested and present these results in new figures (Figure 3—figure supplement 4, Figure 3—figure supplement 5, Figure 3—figure supplement 6, Figure 3—figure supplement 7). We note, however, our dataset is relatively small for reliable estimates. As Kobak et al. pointed out in their 2016 paper, “…at least ~100 neurons were needed to achieve satisfying demixing” for three task parameters (stimulus, choice, and their interactions). “In cases when there are many task parameters of interest, dPCA is likely to be less useful than the more standard parametric single-unit approaches (such as linear regression)”. In our data set, we have ~140 neurons for each region and 7 task parameters (coherence, choice, reward context, and their interactions). Moreover, because we used a decision task with biased reward contexts, the datasets were inherently unbalanced across different combinations of trial types. Because of these constraints, although dPCA provides a helpful characterization of the modulation patterns at the population level, we believe that the multiple linear regression results are likely to be more robust and therefore chose to keep the latter in main figures and the former as supplements.

4) Alternative hypotheses, such as two stage model, pre-stimulus urgency signals, starting point hypothesis, need to be tested and rejected more convincingly.

We apologize for the apparent lack of effort in relating models to the monkeys’ performance. Because we submitted this manuscript as Research Advances, i.e., a follow-up for our previous *eLife* papers that included detailed model-comparison results, we followed the journal’s recommendation of limiting redundant presentation of previous results. We showed in Fan et al., 2018 that pre-stimulus offset alone (i.e., starting point hypothesis) cannot capture the monkeys’ reward-biased performance. A choice-non-specific, pre-stimulus urgency signals cannot capture the asymmetric effects of reward context on the monkeys’ choices. The model we used here is a formulation of the two-stage processing that Diederich and Busemeyer proposed, but for a RT-version task.

Currently, much larger effects in baseline state across a large neural population in FEF is deemphasized compared to the smaller effects seen in the non-choice selective caudate neurons. This might be because the analysis of neural data is based on the conviction that their behavioral model is also the best model for the neural data, but it might be necessary to explore the possibilities beyond their best behavioral model, which might still be wrong. This is also where population analysis (e.g., dPCA) might be helpful. On the one hand, it may be reasonable to only select choice-selective neurons but in doing so we are tossing 70% of the dataset. Only 44 and 36 neurons are now going into your analysis. If the authors decode variables from dPCA with reasonable variance you can just directly look at them in relation to model predictions. It uses the whole dataset and that is the advantage of such an approach.

We apologize for the confusion. In the original manuscript, the analyses were applied to all neurons, regardless of their choice selectivity (e.g., original Figures 3, Figure 7, Figure 7—figure supplement 1, Figure 7—figure supplement 2). Because all accumulation-to-bound models assume that asymmetric bounds-induced choice biases are directional, the choice-selective neurons, if directly contributing to the bound bias, would share the same directionality. In contrast, not-consistently choice-selective neurons do not directly map onto the model assumption. Because these two groups of neurons participate differently in the decision process, we presented their results separately in the original Figure 7 and supplements.

The reviewers’ comments prompted us to more carefully consider how to interpret results from the not-consistently choice-selective neurons. Because accumulation-to-bound models (including the DDM) do not have predictions for this type of neurons, we have decided that our original congruency-based interpretation was not valid. We now present the results from these neurons as supplement to show that they can potentially participate in the decision process (Figure 6—figure supplement 1 and Figure 7—figure supplement 2), but do not assign them any functional roles.

Please see our response above regarding dPCA.

5) The regression analysis used in this analysis shown in Figure 6 was based on the rate of change in firing rate (not the firing rate itself), but how the slope was calculated was not explicitly explained. The details of this should be given in the Materials and methods. For example, to cleanly separate the effect related to drift rate, it might be necessary to remove the contribution from the changes in the baseline firing rate, and simply calculating the slope of the regression model applied to different bins of the neural activity during the 200 ms window might not be sufficient for this.

We apologize for not explaining the method for measuring the slope of change in firing rates. We have added the following text:

“Measuring the slope of change in firing rates

Only correct trials were included for this analysis. Spike trains were aligned to motion onset and grouped by coherence x reward context combinations. The average firing rates were computed for each combination, truncated at median RT for the combination, and convolved with a Gaussian kernel (σ = 20 ms). The slope of change was measured from 200 ms running windows (in 20 ms steps) of the smoothed firing rates for each combination, using a linear regression with time as the independent variable. For each running window, a multiple linear regression was performed, using coherence, reward context, and their interaction as the independent variable and the slopes of change as the dependent variable. Significance for individual regressors was assessed using t-test (criterion: p=0.05).”

With this method, contributions of changes in baseline firing rates were excluded.

Also, isn't it possible that at least some of the effects illustrated in Figure 6C are mediated by the changes in neural activity related to RT? If so and if the contribution of RT is not controlled for, how does this affect the interpretation?

Because the drift rate controls RT, a neural correlate of drift-rate bias is necessarily related to RT. Accordingly, removing the correlation between neural activity and modulations of RT would be equivalent to removing contributions of the drift-rate bias, making it impossible to identify neural correlates of these biases.

6) To distinguish between different scenarios depicted in Figure 4, the authors applied a regression model that relates to the slope of firing rate to coherence, reward context, and their interaction. However, since this model includes the interaction term, the ratio between the regression coefficients for the two main terms is not the most appropriate quantity to test the scenario in Figure 4B. An alternative and simpler method might be to examine the coefficient for coherence and the difference in the average slope in the two reward contexts.

We apologize for not making our rationale clear. In the DDM, the slope of change in the decision variable is governed by a scaling factor (*k*), the actual evidence strength (Coh), and a bias in the momentary evidence (drift-rate bias, *me*). The drift rate for a given coherence level is:

Drift(pref − small − reward blocks) = k0×(Coh+me0) - Eq. 1

Drift(pref − large − reward blocks) = (k0+krew)×(Coh+me0+merew), - Eq. 2

In this formulation,k0 and me0 represent reward context-independent baseline values of the scaling factor and drift-rate bias, and krew and merew represent corresponding reward context-dependent changes, respectively. 𝑚𝑒_𝑟𝑒𝑤_ corresponds to the behaviorally estimated reward-driven drift-rate bias. If a neuron’s firing rate faithfully follows the decision variable in the DDM,

Slope of firing rate =k0×me0+k0×Coh+krew×Coh×Irew+(k0×merew+krew×me0+krew×merew)×Irew - Eq. 3

Where Irew = {0 for pref-small-reward blocks, 1 for pref-large-reward blocks}. The ratio between the two main regressors (i.e., the fourth divided by the second term in Eq. 3) is thus

Coefficient ratio (reward contextcoherence)=merew+(me0+merew)×krew/k0 - Eq. 4

Because k_rew_/ k_0_ tended to be much less than 1 (mean absolute value across all sessions = 0.14), this coefficient ratio is close to the behavioral drift-rate bias (me_rew_) across sessions. In comparison, the difference in the average slope,krew×Coh¯+(k0+krew)×merew+krew×me0,has a more complex relationship with me_rew_ that depends on the values of the krew,k0,me0, and average coherence values Coh¯ that differed among the monkeys. This complex relationship can be seen in the example in Figure 6A and B. We thus consider the coefficient ratio we used (Eq. 4) to be appropriate for testing for a link between neural activity and the behaviorally estimated drift-rate biases.

For example, although the example neuron in Figure 6A might have a significant effect of reward context (in terms of intercept), the effect of reward context varies with coherence effect and reverses for high coherence, which is not consistent with the pattern expected from the biased drift model (Figure 4B).

We agree with the reviewers that the significant modulation by reward context-coherence interaction in the example neuron was not expected from a DDM with only biased drift rates. This difference supports our interpretation that the FEF activity does not fully follow the trajectory of a decision variable in the DDM. We added a reference to this observation in the Discussion.

“For our task, neither FEF nor caudate activity represented the full, latent decision variable as predicted in the DDM framework. For example, in addition to the disconnect between bound bias and reward context-modulated baseline activity, the example FEF neuron in Figure 6A showed a strong modulation by the coherence-reward context interaction, which was not predicted by the DDM.”

In addition, the ratio between the two regression coefficients might produce unreliable results, because they are disproportionally influenced by the denominator (log transformation might be appropriate).

We use the ratio of coefficients for reward context and coherence to control for the variations in firing ranges during motion viewing among neurons. To guard against exactly the reviewers’ concern, we performed this analysis for only neurons with significant coherence modulation, i.e., the denominator was not near zero.

The negative ratios shown for some neurons are also difficult to interpret.

We agree with the reviewers that the negative ratios are not compatible with neural correlates of decision variable in the DDM. These neurons are clearly in the minority in our samples for both regions. In theory, the negative ratio means that a rise in activity is greater when the coherence is higher and when the monkey chose the small-reward target. In a separate, unpublished study, we found that a choice-confidence signal can predict such patterns for our task. Choice confidence is expected to be higher with stronger evidence. Less intuitively, a monkey would only choose the small-reward target when the confidence is high (i.e., confidence is higher on average for small-reward choice trials). A combination of these two effects could lead to negative ratios between coefficients of reward context and coherence for signals related to choice confidence. In the interest of keeping this manuscript focused on model-predicted decision variable-related signals, we added the following text in Discussion to raise this possibility without going into more details:

“Besides decision formation, both FEF and the caudate nucleus have other hypothesized decision-related roles, including performance monitoring (Ding and Gold, 2010, 2012c, 2013; Teichert et al., 2014; Yanike and Ferrera, 2014a). As we showed, a substantial fraction of reward context-modulated neural activity were present in neurons that were not consistently selective for choice. These activity patterns were sensitive to reward biases in drift rates during motion viewing (Figure 6—figure supplement 1) or in the baseline firing before motion onset and saccade onset (Figure 7—figure supplement 2). In addition, some choice-selective neurons showed negative ratios of reward context and coherence coefficients (Figure 6D), which are not consistent with the decision variable predicted by the DDM but could reflect a choice confidence signal instead. It would be important to investigate further the exact functional roles of these activity patterns for solving decision-making tasks.”

7) The dissociation between Figure 6C and D is based on finding a significant correlation in C, but not in D. The result would be strengthened by being able to show that there is a statistical difference between one particular parameter that can be estimated for both FEF and caudate neurons. For example, is it possible to estimate both linear regression slopes and to show that they are statistically different?

As mentioned above, after including monkey A’s data, there was a significant negative correlation for caudate neurons, in contrast to the significant positive correlation for FEF neurons. Given the opposite signs, a statistical test is no longer necessary.

8) To test whether reward-related changes in neural activity was related to the variability in the bound height, they focused on the consistency in the signs of the regression coefficients for reward context and bias in the bound heights (Figure 7D). However, these coefficients are negative for the majority of the neurons, so they do not address the question of whether the variability in these two measures are correlated across sessions (for example, what was the correlation coefficient for the data shown in Figure 7D?).

As mentioned above, this figure is no longer presented. But the reviewers’ comment still applies to analysis of potentially bound asymmetry-related neural signals. Because the value of the reward-context coefficient in our regression analysis depends on a neuron’s firing range, a (lack of) correlation between the coefficient and the behaviorally measured bias cannot be interpreted. Ideally, the bound bias-related neural activity should be normalized to the true value of the bound-related firing rate. However, the latter cannot be measured in practice, particularly because the final bound can change with reward context.

This is an admittedly difficult challenge. Even including trials with equal reward would not necessarily address this challenge, because monkeys can change their total bound heights between equal and asymmetric reward contexts. This is also different from analysis of activity related to drift-rate bias, where the coherence modulation in the same neuron can be used for normalization.

9) The authors have used two regression models to examine the activity of FEF and CD neurons, one including coherence (sensory variable), and the other including RT (motor variable). Since the effects of these variables were modeled separately for each choice (e.g., ipsi vs. contra), they are correlated with each other, but in principle, this should not prevent the authors from including both of these regressors in the same model. This would be preferred, because it is likely that these two factors still influence the activity in either or both of these structures differently.

Because RT depends on reward context and coherence in complicated, non-linear ways, we do not believe that regression results with all three parameters are easily interpretable.

Also, when the effects of coherence and RT were summarized in Figure 3, it would be better to correct them for multiple comparisons.

Our interpretation of the fraction of neurons is based on chance levels corrected for multiple comparisons (e.g., Figure 3A caption: Dashed lines: chance level, adjusted for the number of comparisons. Filled circles: the fraction as significantly greater than chance level (Chi-square test, p<0.05/72 (8 epochs x 9 comparisons).”.

In addition, the reward context was varied across blocks, which would introduce serial correlation in both independent and dependent variables of the regression model, making a standard t-test no longer appropriate (c.f., Elber-Dorozko and Loewenstein, 2018).

We agree with the reviewers that serial correlation due to slow fluctuations in neural activity should be considered for blocked designs. The potential effects of serial correlation are expected to be similar for all time points within the short time scale of a trial. Because our analysis were performed in multiple epochs and time windows (Figure 3) and still showed strong dependence on time/epoch, we did not consider this to be a major caveat. The comment prompted us to perform the permutation test as suggested by Elber-Dorozko and Loewenstein. Now only neurons that passed the p=0.05 cutoff for the standard t-test *and* p=0.05 for the permutation test are identified as significant in Figure 3A-D and Figure 8. We note that applying this additional criterion did not qualitatively change our results.

[Editors' note: further revisions were suggested prior to acceptance, as described below.]

This is a significant study that advances our understanding on the functional specialization of the frontal eye field and caudate nucleus during a perceptual decision making with asymmetric reward outcomes. The reviewers appreciated the extensive amount of work performed by the authors to address their previous concerns, but have one important remaining concern. In particular, the authors are concerned that the relationship between the behavioral bias in drift and the neural effects of reward context seen in the FEF (Figure 6) might be driven by the results from one animal (monkey A). In addition, it is possible that the results might be due to the individual differences among different monkeys rather than the session-by-session (or neuron-by-neuron) changes in behaviors (i.e., Simpson's paradox). Therefore, it is strongly suggested that the authors should analyze and report the results separately for individual animals, or at least repeat this analysis after excluding the results from monkey A.

We followed the reviewers’ suggestion and performed the correlation analysis separately for the three monkeys (Figure 6—figure supplement 2, also see below for your convenience). As expected, given the smaller sample sizes, none of the per-monkey results was statistically significant (monkey C had the highest correlation coefficient, then monkey A, with monkey F’s below zero). We also repeated the analysis with Spearman’s correlation, which tests for a monotonic (not necessarily linear) relationship, and found that monkey C showed a positive coefficient of 0.9 (p=0.037), monkey A showed a non-significant positive coefficient of 0.14 (p=0.66), and monkey F showed a coefficient of -0.1 (p=0.87). These results suggest that the effect we reported in the main figure was not driven by only monkey A or an artifact due to Simpson’s paradox. We have added the following text to clarify our interpretation of these across- (not within-) monkey findings: “As expected given the smaller sample sizes, none of the per-monkey results was statistically significant (Figure 6—figure supplement 2). These results indicated a close relationship between FEF neurons and the neural implementation of reward biases in drift rates assessed across monkeys.”

We hope these new results and explanation address the reviewers’ concern. In addition, because of the somewhat complicated nature of the results that do not necessarily support the idea of “complementary” roles of FEF and caudate, we changed the title to “Frontal eye field and caudate neurons make different contributions to reward-biased perceptual decisions”.